# Linear Bandits with Memory

**Giulia Clerici**                                                                                         *giulia.clerici@unimi.it*
*Department of Computer Science, University of Milan, Italy*

**Pierre Laforgue**                                                                                     *pierre.laforgue1@gmail.com*
*Department of Computer Science, University of Milan, Italy*

**Nicolò Cesa-Bianchi**                                                                          *nicolo.cesa-bianchi@unimi.it*
*Department of Computer Science, University of Milan, Italy*
*DEIB, Politecnico di Milano, Italy*

**Reviewed on OpenReview:** *https://openreview.net/forum?id=CrpDwMFgxr*

## Abstract

Nonstationary phenomena, such as satiation effects in recommendations, have mostly been modeled using bandits with finitely many arms. However, the richer action space provided by linear bandits is often preferred in practice. In this work, we introduce a novel nonstationary linear bandit model, where current rewards are influenced by the learner's past actions in a fixed-size window. Our model, which recovers stationary linear bandits as a special case, leverages two parameters: the window size $m \geq 0$, and an exponent $\gamma$ that captures the rotting ($\gamma < 0$) or rising ($\gamma > 0$) nature of the phenomenon. When both $m$ and $\gamma$ are known, we propose and analyze a variant of OFUL which minimizes regret against cyclic policies. By choosing the cycle length so as to trade-off approximation and estimation errors, we then prove a bound of order $\sqrt{d} \, (m+1)^{\frac{1}{2}+\max\{\gamma,0\}} \, T^{3/4}$ (ignoring log factors) on the regret against the optimal sequence of actions, where $T$ is the horizon and $d$ is the dimension of the linear action space. Through a bandit model selection approach, our results are then extended to the case where both $m$ and $\gamma$ are unknown. Finally, we complement our theoretical results with experiments comparing our approach to natural baselines.

## 1 Introduction

Many real-world problems are naturally modeled by stochastic linear bandits, where actions belong to a linear space, and the learner obtains rewards whose expectations are linear functions of the chosen action (see e.g., Lattimore & Szepesvári (2020)). Formally, at each time step $t$ the expected reward is $r_t = \langle a_t, \theta^* \rangle$, where $a_t \in \mathbb{R}^d$ is the chosen action and $\theta^* \in \mathbb{R}^d$ is a fixed and unknown parameter to be estimated. In a song recommendation problem, for instance, the possible actions are the songs from the catalogue, usually represented by their feature vectors (Deshpande & Montanari, 2012; Korkut & Li, 2021; Ghoorchian & Maghsudi, 2022). The linear reward $r_t$ (i.e., the user satisfaction) measures how well the song $a_t$ picked by the learner matches the (unknown) preferences of the user, represented by $\theta^*$. However, this model fails to capture a key aspect, i.e., the nonstationarity of the users' preferences. For example, user satiation with respect to the recommended items is a typical phenomenon in this context (Kapoor et al., 2015; Kunaver & Požrl, 2017), as studied in rotting bandits (Bouneffouf & Féraud, 2016). Indeed, identifying the favorite song of a user (i.e., the vector $a$ in the action set that maximizes $\langle a, \theta^* \rangle$) only partly solves the recommendation problem, as suggesting this song repeatedly is not meaningful in the long run (Kovacs et al., 2018; Schedl et al., 2018). But satiation is far from being the only nonstationary phenomenon observed in practice. In algorithmic selection for instance, one must choose among a pool of algorithms the one that is going to get the next chunk of resources (e.g., CPU time or samples). In this case, we expect the quality of the solution found by each algorithm to increase as the algorithm gets selected. This model, known as rising bandits,

has been studied in deterministic (Heidari et al., 2016; Li et al., 2020) and stochastic (Metelli et al., 2022) settings.

Nonstationarity in bandits, which has been mostly studied in the case of finitely many arms, appears to be significantly more intricate to analyze in a linear bandit framework due to the structure of the action space. For instance, rotting bandits (Bouneffouf & Féraud, 2016) or rested rising bandits (Metelli et al., 2022) assume that the expected reward of an arm is fully determined by the number of times this arm has been pulled in the past. In the linear case, on the contrary, one would expect nontrivial cross-arm effects. Listening to rock songs should affect the future interest in rock songs, but also to a minor extent that in folk music, as the two genres are related. On the other side, it also seems reasonable that a folk rock song does not increase rock satiation as much as a pure rock song. Hence, a principled way to model nonstationarity in linear environments is needed.

In this work, we introduce a novel linear bandit framework that allows to model complex nonstationary behaviors in an infinite and structured space of actions. More specifically, the nonstationarity is captured by a matrix, determined by the past actions of the learner and affecting the expected reward of future actions. Formally, the expected reward at time step $t$ becomes $r_t = \langle a_t, A_{t-1}\theta^* \rangle$, where $A_{t-1} = A(a_{t-1}, \ldots, a_{t-m}) = \left(A_0 + \sum_{s=1}^{m} a_{t-s}a_{t-s}^\top\right)^\gamma \in \mathbb{R}^{d \times d}$. Here, $A_0$ is some initial symmetric and positive semidefinite matrix. Typically, $A_0$ is chosen to be the identity $I_d$, which we refer to as the isotropic initialization. The memory size $m \geq 0$ controls the range of past actions having an influence, while the exponent $\gamma \in \mathbb{R}$ quantifies their impact. A positive $\gamma$ corresponds to a rising behavior, and a negative $\gamma$ to a rotting one — two established scenarios in the bandit literature. In the rotting setting, playing action $a$ at time $t$ decreases the expected reward of $a$ at time $t+1$. Hence, solving this problem requires long-term planning, and playing repeatedly $\theta^*$ may not be optimal. Instead, in a rising scenario with isotropic initialization, an optimal action played (and thus boosted) at time $t$ remains optimal at time $t+1$. Although optimal policies are stationary in this case, note that such problems are intrinsically difficult as the learner is penalized twice: for not choosing a good action at the present time, but also at future time steps, for not having boosted the right action. We highlight that our approach is able to cope simultaneously with these two different scenarios. Finally, note that our model recovers stationary linear bandits as a special case when $\gamma = 0$ (or $m = 0$ and $A_0 = I_d$).

We start by focusing on cyclic policies, and show that they provide a reasonable approximation to the optimal policy (which may not be cyclic) while being easier to learn. When $m$ and $\gamma$ are known, estimating the best block of fixed length reduces to a stationary problem, that we solve using a block variant of OFUL (Abbasi-Yadkori et al., 2011). When $m = 0$, our variant recovers the regret bound $\mathcal{O}\left(d\sqrt{T}\right)$ of OFUL up to log factors. We then optimize the block length in order to balance the approximation and estimation errors, and obtain a bound on the regret against the optimal sequence of actions in hindsight of order $\sqrt{d}\,(m+1)^{\frac{1}{2}+\max\{\gamma,0\}}T^{3/4}$ (ignoring log factors) for all $T \geq (md)^2$. Finally, we extend our analysis to the case when $m$ and $\gamma$ are both unknown. For this case, we prove regret bounds via an extension of the bandit model selection approach of Cutkosky et al. (2020). Empirically, our approach is shown to outperform natural baselines, such as the oracle greedy strategy (playing the action with the best instantaneous expected reward) and a naive block learning approach. Our experimental results also include misspecified settings, where we learn $\theta^*$ and simultaneously either $m$ or $\gamma$.

**Contributions.**

- We introduce a new bandit framework to model nonstationary effects in linear action spaces. Our model generalizes stationary linear bandits, whose bound we recover as a special case.
- We propose an OFUL-based algorithm achieving sublinear regret against the best sequence of actions by learning cyclic policies and balancing estimation and approximation errors.
- We use a bandit model selection approach to learn the system's parameters $m$ and $\gamma$.
- Empirically, our algorithm outperforms natural baselines in both rotting and rising settings.

**Related works.** Stochastic linear bandits, which were introduced two decades ago (Abe & Long, 1999; Auer, 2002), are typically addressed using algorithms based on ellipsoidal confidence sets (Dani et al., 2008; Rusmevichientong & Tsitsiklis, 2010; Abbasi-Yadkori et al., 2011). Nonstationary bandits have been mainly studied in the case of finitely many arms. Among the most studied models, there are rested (Gittins, 1979;

Gittins et al., 2011) and restless (Whittle, 1988; Ortner et al., 2012; Tekin & Liu, 2012) bandits, rotting bandits (Bouneffouf & Féraud, 2016; Heidari et al., 2016; Cortes et al., 2017; Levine et al., 2017; Seznec et al., 2019), bandits with rewards depending on arm delays (Kleinberg & Immorlica, 2018; Cella & Cesa-Bianchi, 2020; Simchi-Levi et al., 2021; Laforgue et al., 2022), blocking and rebounding bandits (Basu et al., 2019; Leqi et al., 2021), and rising bandits (Li et al., 2020; Metelli et al., 2022).

The $d$-step lookahead regret of Pike-Burke & Grunewalder (2019) is similar to our regret against the best cyclic policy. However, while the lookahead oracle selects the best block based on the learner's current state, our oracle is defined independently of the learner's action. In this respect, our work investigates a policy regret version of the lookahead regret. Some works have also considered nonstationary bandit frameworks, where the unknown parameter $\theta^*$ is then replaced by a sequence of vectors $\theta_t^*$ that evolves over time. Standard assumptions then stipulate that $\theta_t^*$ is piecewise stationary, with a fixed number of change points (Bouneffouf et al., 2017; Wu et al., 2018; Auer et al., 2019; Chen et al., 2019; Di Benedetto et al., 2020; Xu et al., 2020; Li et al., 2021), or that the variation budget $\sum_{t \leq T} \|\theta_t^* - \theta_{t-1}^*\|$ is bounded (Besbes et al., 2014; Karnin & Anava, 2016; Luo et al., 2018; Cheung et al., 2019; Russac et al., 2019; 2020; Kim & Tewari, 2020; Zhao et al., 2020). See also Mueller et al. (2019) for an application of linear bandits to nonstationary dynamic pricing. In addition to these assumptions, we highlight that the above works are fundamentally different from ours, as the evolution of $\theta_t^*$ is oblivious to the actions taken by the learner. This removes any need for long-term planning and puts the focus on the dynamic regret, where the algorithm's performance is compared to the rewards which one could obtain by picking $a_t$ according to $\theta_t^*$. Finally, note that nonstationarity in linear bandit environments may also be tackled using Gaussian Processes (Faury et al., 2021; Deng et al., 2022).

We note that the idea of combining linear and rotting bandits was already discussed in Seznec (2020, Section 4.7), where the author provides some evidences on the intrinsic difficulty to do so. There, the author proposes an extension of rotting bandits to linear spaces of actions by summing along the different dimensions the projections of the past actions. It is however proved that such a model cannot be learned. Indeed, it is possible to exhibit an instance of this linear rotting problem for which any policy suffers linear regret. On the contrary, our analysis in Section 3 shows that our model (based instead on the covariance matrix of the past actions) is learnable. The price we pay for ensuring learnability is that our model does not capture the $K$-armed rotting bandit setting in its full generality, see Example 2 for more details.

**Notation.** $\mathcal{B}_d$ denotes the Euclidean unit ball, $0_d$ and $(e_k)_{k \leq d}$ the zero and standard basis in $\mathbb{R}^d$, $I_d \in \mathbb{R}^{d \times d}$ the identity matrix, $\|M\|_*$ the operator norm of $M$, and $\gamma^+ = \max(\gamma, 0)$ for any $\gamma \in \mathbb{R}$. Bold characters refer to block objects, and $\widetilde{\mathcal{O}}$ is used when neglecting logarithmic factors.

## 2 Model

In this section, we introduce our model of *linear bandits with memory* (LBM in short). LBMs strictly generalize stationary linear bandits, and also recover some nonstationary bandit models with finitely many arms as special cases. The learning setup is as follows. At each time step $t = 1, 2, \ldots$ the learner picks an action $a_t$ from a (possibly infinite) set of actions $\mathcal{A} \subset \mathcal{B}_d$, and receives a stochastic reward $y_t$. Similarly to linear models, we assume that the expected reward is a linear function of some unknown vector $\theta^* \in \mathcal{B}_d$. In contrast to stationary models, however, the expected reward at time $t$ is also influenced by the choice of previous actions of the learner. Mathematically, this is captured by the correlation matrix $\sum_{s=1}^m a_{t-s} a_{t-s}^\top$, where $m$ measures how far in the past actions can influence the current reward.[1] Finally, in order to model the type (rising or rotting) of behavior and its strength, we use a positive or negative exponent $\gamma$. This results in the following formula for the reward at time $t$,

$$y_t = \langle a_t, A(a_{t-m}, \ldots, a_{t-1}) \theta^* \rangle + \eta_t, \tag{1}$$

---

[1] Using a different analysis, one could replace our fixed-size window with exponentially decaying discount factors. However, while these factors are typically treated as fixed model parameters, our analysis shows how to learn the best $m$ (see Section 3.3).

where $\eta_t$ is a 1-sub-Gaussian random variable independent of the actions of the learner, and

$$A(a_1, \ldots, a_m) = \left( A_0 + \sum_{s=1}^{m} a_s a_s^\top \right)^\gamma.$$  (2)

Equations (1) and (2) define a model which strictly generalizes standard linear bandits, recovered for $\gamma = 0$. The choice of the covariance matrix for (2) is intuitive, as it stores the previously played actions and thus naturally encodes the directions where satiation or excitation occurs through its eigenvectors, see Figure 1. For simplicity, in the rest of the paper we use the abbreviation $A_{t-1} = A(a_{t-m}, \ldots, a_{t-1})$ and refer to it as the *memory matrix*. Conventionally, we set $a_{1-m} = a_{2-m} = \ldots = a_0 = 0_d$ and choose $A_0 = I_d$ unless otherwise stated. Note that parameters $m$ and $\gamma$ have the twofold advantage of making the model general enough to account for both rotting ($\gamma < 0$) and rising ($\gamma > 0$) scenarios while being simple enough to be learned simultaneously with $\theta^*$, see Section 3.3. Note also that at any time step $t$ the expected reward $r_t = \mathbb{E}[y_t]$ satisfies $|r_t| \leq \|A_{t-1}\|_*$. Given a horizon $T \in \mathbb{N}$, the learner aims at maximizing the expected sum of rewards obtained over the $T$ interaction rounds. The performance is measured against the best sequence of actions over the $T$ rounds, i.e., through the regret

$$\sum_{t=1}^{T} r_t^* - \mathbb{E}\left[ \sum_{t=1}^{T} y_t \right],$$

where $r_t^* = \left\langle a_t^*, A(a_{t-m}^*, \ldots, a_{t-1}^*) \theta^* \right\rangle$ and $(a_t^*)_{t \geq 1}$ is the optimal sequence of actions, i.e., the sequence maximizing the expected sum of rewards obtained over the horizon $T$

$$a_1^*, \ldots, a_T^* = \arg\max_{a_1, \ldots, a_T \in \mathcal{A}} \sum_{t=1}^{T} \left\langle a_t, A(a_{t-m}, \ldots, a_{t-1}) \theta^* \right\rangle.$$  (3)

Throughout the paper, we use OPT to denote $\sum_t r_t^*$ whenever the horizon $T$ is understood from the context. Note that a LBM is fully characterized by: the action set $\mathcal{A}$, the parameter $\theta^*$, the memory size $m$, and the exponent $\gamma$. As shown in the following examples, LBMs fully generalize (stationary) linear bandits, and allow to partially recover rotting/rising rested bandits in the limit $m \to \infty$.

**Example 1 (Stationary linear bandits)** *Consider a linear bandit model, defined by an action set $\mathcal{A} \subset \mathcal{B}_d$ and $\theta^* \in \mathcal{B}_d$. This is equivalent to a LBM with the same $\mathcal{A}$ and $\theta^*$, and memory matrix $A$ such that $A(a_1, \ldots, a_m) = I_d$ for any $a_1, \ldots, a_m \in \mathcal{A}^m$, i.e., when $m = 0$ or $\gamma = 0$.*

**Example 2 (Rotting and rising rested bandits)** *In rotting (Levine et al., 2017; Seznec et al., 2019) or rising (Metelli et al., 2022) rested bandits, the expected reward of an arm $k$ at time step $t$ is fully determined by the number $n_k(t)$ of times arm $k$ has been played before time $t$. Formally, each arm is equipped with a function $\mu_k$ such that the expected reward at time $t$ is given by $\mu_k(n_k(t))$. In particular, requiring all the $\mu_k$ to be nonincreasing corresponds to the rotting bandits model, and requiring all the $\mu_k$ to be nondecreasing corresponds to the rested rising bandits model. Now, let $d = K$, $\mathcal{A} = (e_k)_{1 \leq k \leq K}$, $\theta^* = (1/\sqrt{K}, \ldots, 1/\sqrt{K})$, and $m \to \infty^2$. By the definition of $A$, see (2), and the orthogonality of the actions, it is easy to check that the expected reward of playing action $e_k$ at time step $t$ is given by $(1 + n_k(t))^\gamma/\sqrt{K}$. When $\gamma \leq 0$, this is a nonincreasing function of $n_k(t)$, and we recover rotting rested bandits. Conversely, when $\gamma \geq 0$, we recover rising rested bandits. We note however that the class of decreasing (respectively increasing) functions we can consider is restricted to the set of monomials of the form $n \mapsto (1 + n)^\gamma/\sqrt{K}$, for $\gamma \leq 0$ (respectively $\gamma \geq 0$). Extending it to generic polynomials is clearly possible, although it requires more computations in the model selection phase, see Remark 4 and Section 3.3.*

*Although rotting and rising bandits require infinite memory, we argue on both practical and theoretical grounds that in our setting a finite value of $m$ is preferable. First, in many applications it is reasonable to assume that the effect of past actions will vanish at some point. For example, listening to a song now does not affect how much we will enjoy the same song in a distant enough future. Second, permanent effects may trivialize*

---

[2]In the next paragraph, however, we explain why a bounded memory $m$ is preferable within our model.

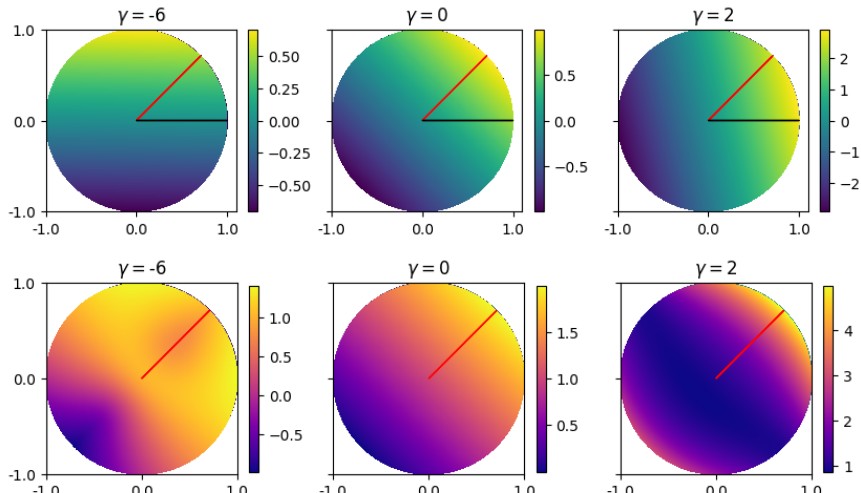

Figure 1: In the top pane, we plot the effect of the memory matrix (2) on the action space for $d = 2$, $m = 1$, and $\gamma \in \{-6, 0, 2\}$. The red arrow is $\theta^*$ and the black arrow is action $a_{t-1}$. The color level indicates the value of the instantaneous expected reward of any action $a_t$ (point on the disk). When $\gamma = -6$, the rotting effect is so powerful that the optimal action $a_t$ is orthogonal to $a_{t-1}$. When $\gamma = 0$, the optimal action remains $\theta^*$, independently of $a_{t-1}$. For $\gamma = 2$, the optimal action is shifted between $\theta^*$ and $a_{t-1}$. However, the top plot does not show that playing constantly $\theta^*$ is not the optimal policy. In the bottom pane, we consider horizon $T = 2$, with the same choices of parameters. For a given action $a_1$, since $T = 2$, it is possible to determine the best possible next action $a_2$. The color now indicates the sum of expected rewards as a function of the initial action $a_1$ (point on the disk). For $\gamma = -6$, we clearly see that playing $\theta^*$ is not optimal anymore. On the other side, it shows that not playing $\theta^*$ is more harmful when $\gamma = 2$ than when $\gamma = 0$.

*the problem on the theoretical side: consider $m \to \infty$ and $\gamma \leq -1/2$, then for any sequence of actions $(a_t)_{t \geq 1}$ we have*

$$\textstyle\sum_{t=1}^{T} \langle a_t, A_{t-1}\theta^* \rangle \leq \sum_{t=1}^{T} \left\| A_{t-1}a_t \right\|_2 \leq \sqrt{T \sum_{t=1}^{T} \left\| A_{t-1}a_t \right\|_2^2} \leq \sqrt{2dT \log(1 + T/d)} \coloneqq B_T \,,$$

*where we have used the elliptical potential lemma (Lattimore & Szepesvári, 2020, Lemma 19.4). Hence, as soon as $\gamma \leq -1/2$, we have $\mathrm{OPT} \leq B_T$, and the trivial strategy consistently playing $0$ enjoys a small regret $B_T$. Conversely, consider $\gamma \geq 0$. The strategy consistently playing $\theta^*$ achieves, after $t$ rounds, an instantaneous reward of $(1 + t)^\gamma$, which is diverging for $\gamma \geq 1$. This is not realistic in most application and, incidentally, violates the concave payoffs assumption (Metelli et al., 2022, Assumption 3.2). Therefore, although considering $m = +\infty$ may look attractive at first sight, it actually fails to adequately model song satiation, and restricts the range of relevant $\gamma$ from $\mathbb{R}$ to $(-1/2, 1)$. Instead, focusing on finite memory $m$ yields more interesting problems, although it prevents a full generalization of rotting bandits with finitely many arms. We note however that when $m < \infty$, the spirit of rotting (resp., rising) bandits is still preserved, as playing an action does decrease (resp., increase) its efficiency for the next pulls (within the time window), see also Figure 1.*

A naive approach to learning LBM is to neglect nonstationarity. Assuming that $\theta^*$ is known, one may then play at time $t$ the action $a_t^{\mathrm{greedy}} = \arg\max_{a \in \mathcal{A}} \langle a, A_{t-1}\theta^* \rangle$. Although this strategy, which we refer to as *oracle greedy*, may be optimal in some cases (e.g., in rising isotropic settings, see Heidari et al. (2016, Section 3.1) and Metelli et al. (2022, Theorem 4.1) for discussions in the $K$-armed case), we highlight that it may also be arbitrarily bad, as stated in the next proposition.

**Proposition 1** *The oracle greedy strategy, which plays $a_t^{\mathrm{greedy}} = \arg\max_{a \in \mathcal{A}} \langle a, A_{t-1}\theta^* \rangle$ at time step $t$, can suffer linear regret, both in rotting or rising scenarios.*

Hence, one must resort to more sophisticated strategies, which may include long-term planning. Before describing our approach in the next section, we conclude the model exposition by highlighting that LBMs may also be generalized to contextual bandits (Lattimore & Szepesvári, 2020).

**Remark 1 (Contextual bandits)** *In contextual bandits, at each time step $t$ the learner is provided a context $c_t$ (e.g., data about a user). The learner then picks an action $a_t \in \mathcal{A}$ (based on $c_t$), and receives a reward whose expectation depends linearly on the vector $\psi(c_t, a_t) \in \mathbb{R}^d$, where $\psi$ is a known feature map. Note that it is equivalent to have the learner playing actions $a_t \in \mathbb{R}^d$ that belong to a subset $\mathcal{A}_t = \{\psi(c_t, a) \in \mathbb{R}^d : a \in \mathcal{A}\}$. The analysis developed in Section 3 still holds true when $\mathcal{A}_t$ depends on $t$, and can thus be generalized to contextual bandits with memory.*

## 3 Regret Analysis

In this section, we introduce and analyze OFUL-memory (Algorithm 1) for learning LBMs. We first observe that for every block length there exists a cyclic policy providing a reasonable approximation to the optimal policy (Proposition 2) that cannot be improved in general, see Proposition 3. Learning the optimal block in the cyclic policy then reduces to a stationary linear bandit problem that can be solved by running the OFUL algorithm (Proposition 4). This approach is however wasteful, as it estimates a concatenated model whose dimension scales with the block length. We thus propose a refined algorithm leveraging the structure of the concatenated model, and show that it enjoys a better regret bound. We then tune the block length to trade-off estimation and approximation errors (Theorem 1). Since the optimal block length depends on the memory size $m$, which may be unknown in practice, we finally wrap our algorithm with a bandit model selection algorithm that is shown to preserve regret guarantees (Corollary 1). Throughout the analysis, we assume for simplicity that the horizon $T$ is always divisible by the block length considered. Finally, note that all technical proofs are relegated to the Appendix (Proposition 4 and Theorem 1 being proved with high probability while stated in expectation in the main body for simplicity of exposition).

### 3.1 Approximation

In LBMs, finding a block of actions maximizing the sum of expected rewards is not a well-defined problem. Indeed, the rewards also depend on the initial conditions, determined by the $m$ actions preceding the current block. To bypass this issue, we introduce the following proxy reward function. For any $m, L \geq 1$ and any block $\boldsymbol{a} = a_1 \ldots a_{m+L}$ of $m + L$ actions, let

$$\widetilde{r}(\boldsymbol{a}) = \sum_{t=m+1}^{m+L} \langle a_t, A_{t-1}\theta^* \rangle = \sum_{t=m+1}^{m+L} \langle A_{t-1}a_t, \theta^* \rangle . \tag{4}$$

In words, we only consider the expected rewards obtained from the index $m + 1$ onward. Note that actions $a_1 \ldots a_m$ still do play a role in $\widetilde{r}$, as they influence $A_m, \ldots, A_{2m-1}$. The key is that $\widetilde{r}$ is now independent from the initial state, so that

$$\widetilde{\boldsymbol{a}} = \underset{\boldsymbol{a} \in \mathcal{B}_d^{m+L}}{\arg\max} \ \widetilde{r}(\boldsymbol{a}) \tag{5}$$

is well-defined. The next proposition quantifies the approximation error incurred when playing cyclically $\widetilde{\boldsymbol{a}}$ instead of the optimal sequence of actions $(a_t^*)_{t \leq T}$ defined in (3). A critical quantity to establish this result is the maximal (and minimal) instantaneous reward one can obtain. To this end, we introduce the notation $R = \sup_{a_1, \ldots, a_{m+1} \in \mathcal{A}} |\langle a_{m+1}, A(a_1, \ldots, a_m)\theta^* \rangle|$. Note that in (8) we provide a bound on $R$ in terms of $m$ and $\gamma$. We now state our approximation result, and show that it is tight up to constant.

**Proposition 2** *For any $m, L \geq 1$, let $\widetilde{\boldsymbol{a}}$ be the block of $m + L$ actions defined in (5) and $(\widetilde{r}_t)_{t=1}^T$ be the expected rewards collected when playing cyclically $\widetilde{\boldsymbol{a}}$. We have*

$$\text{OPT} - \sum_{t=1}^T \widetilde{r}_t \leq \frac{2mR}{m+L} T . \tag{6}$$

The dependence on the cycle length $L$ of the right-hand side of (6) is as expected: by increasing $L$, the expected reward of the cyclic policy gets closer to OPT. In addition, note that for $m = 0$ we recover the stationary behaviour. In this case, there are no long-term effects and the performance is oblivious to the block length, so that we recover $\sum_t \widetilde{r}_t = \text{OPT}$ independently of $L$. Next, we show that Proposition 2 is tight up to constants.

**Proposition 3 (Tight approximation)** *For any $m, L \geq 1$ and $\gamma \leq 0$, let $\widetilde{\boldsymbol{a}}$ be the block of $m + L$ actions defined in (5) and $(\widetilde{r}_t)_{t=1}^T$ be the expected rewards collected when playing cyclically $\widetilde{\boldsymbol{a}}$. Then, there exists a choice of $\mathcal{A}$ and $\theta^*$ such that*

$$\text{OPT} - \sum_{t=1}^{T} \widetilde{r}_t \geq \frac{mR}{m+L} T \;. \tag{7}$$

Upper bounds on $R$ are easy to obtain. Let $a_1, \ldots, a_{m+1} \in \mathcal{A}$, and $A_m = A(a_1, \ldots, a_m)$, we have

$$|r_m| = \left| \langle a_{m+1}, A_m \theta^* \rangle \right| \leq \|a_{m+1}\|_2 \, \|A_m \theta^*\|_2 \leq \|A_m\|_* \, \|\theta^*\|_2 \leq (m+1)^{\gamma^+} \;, \tag{8}$$

such that one can take $R = (m+1)^{\gamma^+}$. Note that any other choice of dual norms could have been used to upper bound $\left| \langle a_{m+1}, A_m \theta^* \rangle \right|$, as done in Proposition 3. For simplicity, we restrict ourselves to the Euclidean norm from now on, and use $R = (m+1)^{\gamma^+}$.

**Remark 2 (On the necessity of optimizing over the first actions.)** *We highlight that optimizing over the first $m$ actions in Equation (5) is necessary, as there exists no such "pre-sequence" which is universally optimal. Indeed, let $A_t$ and $A_t'$ be the memory matrices generated by $a_1 \ldots a_{m+L}$ and $a_1' \ldots a_m' a_{m+1} \ldots a_{m+L}$ respectively. It is immediate to check that if the pre-sequence $a_1 \ldots a_m$ is better than $a_1' \ldots a_m'$ with respect to some model $\theta \in \mathbb{R}^d$, i.e., if we have $\sum_{t=m+1}^{m+L} \langle a_t, A_{t-1}\theta \rangle \geq \sum_{t=m+1}^{m+L} \langle a_t, A_{t-1}'\theta \rangle$, then the opposite holds true for $-\theta$. Hence, one cannot determine a priori a good pre-sequence and has to optimize for it.*

## 3.2 Estimation

The next step now consists in building a sequence of blocks with small regret against $\widetilde{\boldsymbol{a}}$. As detailed below, this reduces to a stationary linear bandit problem, with a specific action set. After showing an initial naive solution, we provide a refined approach which exploits the structure of the latent parameter and enjoys improved regret guarantees.

**A naive approach.** We introduce some notation first. Let $\boldsymbol{\theta}^* = (0_d, \ldots, 0_d, \theta^*, \ldots, \theta^*) \in \mathbb{R}^{d(m+L)}$ be the vector concatenating $m$ times $0_d$ and $L$ times $\theta^*$. Inspired by the right-hand side in (4), we introduce the subset of $\mathbb{R}^{d(m+L)}$ composed of the blocks $\boldsymbol{b} = b_1 \ldots b_{m+L}$ whose actions are of the form $b_i = A_{i-1} a_i$ for some block $\boldsymbol{a} \in \mathcal{A}^{m+L}$. Formally, let

$$\mathcal{B} = \left\{ \boldsymbol{b} \in \mathbb{R}^{d(m+L)} : \exists \, \boldsymbol{a} \in \mathcal{A}^{m+L} \text{ such that } \begin{cases} b_i = a_i & 1 \leq i \leq m \\ b_i = A_{i-1} a_i & m+1 \leq i \leq m+L \end{cases} \right\} ,$$

where the $(A_i)_{i=m+1}^{m+L-1}$ are the memory matrices generated from $\boldsymbol{a}$. Equipped with this notation, it is easy to see that for any $\boldsymbol{a} \in \mathcal{A}^{m+L}$ and the corresponding $\boldsymbol{b} \in \mathcal{B}$ we have $\widetilde{r}(\boldsymbol{a}) = \langle \boldsymbol{b}, \boldsymbol{\theta}^* \rangle$. Therefore, estimating $\widetilde{\boldsymbol{b}}$ (the block in $\mathcal{B}$ associated to $\widetilde{\boldsymbol{a}}$) reduces to a standard stationary linear bandit problem in $\mathbb{R}^{d(m+L)}$, with parameter $\boldsymbol{\theta}^*$ and feasible set $\mathcal{B}$. In other words, we have transformed the nonstationarity of the rewards into a constraint on the action set. Running OFUL (Abbasi-Yadkori et al., 2011) then amounts to playing at time step $t = \tau(m + L)$, the block $\boldsymbol{a}_\tau \in \mathcal{A}^{m+L}$, whose associated block $\boldsymbol{b}_\tau$ in $\mathcal{B}$ satisfies

$$\boldsymbol{b}_\tau = \arg\max_{\boldsymbol{b} \in \mathcal{B}} \sup_{\boldsymbol{\theta} \in \mathcal{C}_{\tau-1}} \langle \boldsymbol{b}, \boldsymbol{\theta} \rangle \;, \tag{9}$$

where $\mathcal{C}_\tau = \left\{ \boldsymbol{\theta} \in \mathbb{R}^{d(m+L)} : \left\| \widehat{\boldsymbol{\theta}}_\tau - \boldsymbol{\theta} \right\|_{\boldsymbol{V}_\tau} \leq \beta_\tau(\delta) \right\}$, with $\beta_\tau(\delta)$ defined in Equation (17), $\boldsymbol{V}_\tau = \sum_{\tau'=1}^{\tau} \boldsymbol{b}_{\tau'} \boldsymbol{b}_{\tau'}^\top + \lambda I_{d(m+L)}$, $\boldsymbol{y}_\tau = \sum_{i=m+1}^{m+L} y_{\tau,i}$, using $y_{\tau,i}$ to denote the reward obtained by the $i^{\text{th}}$ action of block $\tau$, and

$\widehat{\boldsymbol{\theta}}_\tau = \boldsymbol{V}_\tau^{-1} \big( \sum_{\tau'=1}^{\tau} \boldsymbol{y}_{\tau'} \boldsymbol{b}_{\tau'} \big)$. Noticing that $\|\boldsymbol{\theta}^*\|_2^2 \leq L$, that for any block $\boldsymbol{b} \in \boldsymbol{\mathcal{B}}$ we have $\|\boldsymbol{b}\|_2^2 \leq m + L(m+1)^{2\gamma^+}$ and $\langle \boldsymbol{\theta}^*, \boldsymbol{b} \rangle \leq L(m+1)^{\gamma^+}$, and adapting the OFUL's analysis, we get the following regret bound.

**Proposition 4** *Let* $\lambda \in [1, d]$, $L \geq m$, *and* $\boldsymbol{a}_\tau$ *be the blocks of actions in* $\mathbb{R}^{d(m+L)}$ *associated to the* $\boldsymbol{b}_\tau$ *defined in* (9). *Then we have*

$$\mathbb{E}\left[ \sum_{\tau=1}^{T/(m+L)} \widetilde{r}(\widetilde{\boldsymbol{a}}) - \widetilde{r}(\boldsymbol{a}_\tau) \right] = \widetilde{\mathcal{O}}\Big( dL^{3/2}(m+1)^{\gamma^+} \sqrt{T} \Big)\,.$$

In the stationary case, i.e., when $m = 0$ and $L = 1$, the block approach coincide with OFUL and we do recover (up to log factors) the $\mathcal{O}(d\sqrt{T})$ bound for standard linear bandits. Note that in Proposition 5 in the Supplementary Material we prove a more general high-probability bound, which also specializes to known results for linear bandits in the stationary case.

**A refined approach.**  Note however that the approach presented above is wasteful. Indeed, while the relevant model to estimate is $\theta^* \in \mathbb{R}^d$, the $\widehat{\boldsymbol{\theta}}_\tau$ are estimators of the concatenated vector $\boldsymbol{\theta}^* \in \mathbb{R}^{d(m+L)}$, with degraded accuracy due to the increased dimension. Similarly, this method only uses the sum of rewards obtained by a block, while finer-grained information is available, namely the rewards obtained by each individual action in the block. Driven by these considerations, let $\boldsymbol{a}_\tau = a_{\tau,1} \ldots a_{\tau,m+L}$ be the block of actions played at block time step $\tau$, $A_{\tau,i-1} = A(a_{\tau,i-m}, \ldots, a_{\tau,i-1})$, and $b_{\tau,i} = A_{\tau,i-1} a_{\tau,i}$ for $i \geq m$. We propose to compute instead

$$\widehat{\theta}_\tau = V_\tau^{-1} \left( \sum_{\tau'=1}^{\tau} \sum_{i=m+1}^{m+L} y_{\tau',i}\, b_{\tau',i} \right), \tag{10}$$

where $V_\tau = \sum_{\tau'=1}^{\tau} \sum_{i=m+1}^{m+L} b_{\tau',i} b_{\tau',i}^\top + \lambda I_d$. In words, $\widehat{\theta}_\tau$ is the standard regularized least square estimator of $\theta^*$ when only the last $L$ rewards of each block of size $m + L$ are considered. Note however that the $\widehat{\theta}_\tau$ are only computed every $m + L$ rounds. Indeed, recall that regret is computed here at the block level, such that at each block time step $\tau$ the learner chooses upfront an entire block to play, preventing from updating the estimates between the individual actions of the block. Following the principle of optimism in the face of uncertainty, a natural strategy then consists in playing

$$\boldsymbol{a}_\tau = \arg\max_{a_{\tau,i} \in \mathcal{A}} \sup_{\theta \in \mathcal{C}_{\tau-1}} \sum_{i=1}^{L} \langle a_{\tau,i}, A_{\tau,i-1}\theta \rangle, \tag{11}$$

where $\mathcal{C}_\tau = \big\{ \theta \in \mathbb{R}^d \colon \big\| \widehat{\theta}_\tau - \theta \big\|_{V_\tau} \leq \beta_\tau(\delta) \big\}$, for some $\beta_\tau(\delta)$ defined in (18). Expressed in terms of $\boldsymbol{b}_\tau$, the estimate (11) corresponds to

$$\boldsymbol{b}_\tau = \arg\max_{\boldsymbol{b} \in \boldsymbol{\mathcal{B}}} \sup_{\boldsymbol{\theta} \in \boldsymbol{\mathcal{D}}_{\tau-1}} \langle \boldsymbol{b}, \boldsymbol{\theta} \rangle, \tag{12}$$

where $\boldsymbol{\mathcal{D}}_\tau = \big\{ \boldsymbol{\theta} \in \mathbb{R}^{d(m+L)} \colon \exists \theta \in \mathcal{C}_\tau \text{ such that } \boldsymbol{\theta} = (0_d, \ldots, 0_d, \theta, \ldots, \theta) \big\}$. In words, this estimate is similar to (9), except that we use the improved confidence set $\boldsymbol{\mathcal{D}}_\tau$ that leverages the structure of $\boldsymbol{\theta}^*$. A dedicated analysis to deal with the fact that the estimates $\widehat{\theta}_\tau$ are not "up to date" for actions inside the block then allows to bound the regret of the sequence $\boldsymbol{a}_\tau$ against the optimal $\widetilde{\boldsymbol{a}}$. Setting the block size $L$ in order to balance this bound with the approximation error of Proposition 2 yields the final regret bound.

**Theorem 1** *Let* $\lambda \in [1, d]$, *and* $\boldsymbol{a}_\tau$ *be the blocks of actions in* $\mathbb{R}^{d(m+L)}$ *defined in* (11). *Then we have*

$$\mathbb{E}\left[ \sum_{\tau=1}^{T/(m+L)} \widetilde{r}(\widetilde{\boldsymbol{a}}) - \widetilde{r}(\boldsymbol{a}_\tau) \right] = \widetilde{\mathcal{O}}\Big( dL(m+1)^{\gamma^+} \sqrt{T} \Big)\,.$$

*Suppose that* $m \geq 1$, $T \geq d^2 m^2 + 1$, *and set* $L = \lceil \sqrt{m/d}\, T^{1/4} \rceil - m$. *Let* $y_t$ *be the rewards collected when playing* $\boldsymbol{a}_\tau$ *as defined in* (11). *Then we have*

$$\mathrm{OPT} - \mathbb{E}\left[ \sum_{t=1}^{T} y_t \right] = \widetilde{\mathcal{O}}\left( \sqrt{d}\, (m+1)^{\frac{1}{2}+\gamma^+}\, T^{3/4} \right)\,.$$

*When $m = 0$ (i.e., in the stationary case), setting $L = 1$ recovers the OFUL bound.*

When comparing the first claim of Theorem 1 to Proposition 4, we note that the dependence in $L$ has been reduced from $L^{3/2}$ to $L$, thanks to the improved confidence sets. Solving the approximation-estimation tradeoff using Proposition 4 would have yielded an overall regret bound of order $d^{2/5}(m+1)^{\frac{3}{5}+\gamma^+} T^{4/5}$, worse than the bound provided by the second claim of Theorem 1. In the stationary case (i.e., for $m = 0$) Theorem 1 recovers the OFUL regret bound and matches the lower bound for stationary linear bandits (Lattimore & Szepesvári, 2020, Theorems 24.1 and 24.2, e.g.), such that our analysis is tight in general (recall that Proposition 3 shows that the control of the approximation error provided by Proposition 2 is optimal up to constants). Finding a lower bound matching Theorem 1 for arbitrary values of $m$ and $\gamma$ remains however an open problem. We highlight that lower bounds for nonstationary bandits are particularly hard to obtain and that most papers on this topic do not prove any, see e.g., Levine et al. (2017); Kleinberg & Immorlica (2018); Pike-Burke & Grunewalder (2019); Cella & Cesa-Bianchi (2020); Metelli et al. (2022).

As we can see from the optimal choice of $L$ in Theorem 1, OFUL-memory requires the knowledge of the horizon $T$, the memory size $m$, and the exponent $\gamma$, which might all be unknown in practice. If adaptation to $T$ can be achieved by using the doubling trick, adaptation to $m$ and $\gamma$ is more involved. In the next section, we show that OFUL-memory can be wrapped by a model selection algorithm to learn $m$ and $\gamma$. Before turning to this problem, we state a few remarks.

**Remark 3 (An over-optimistic variant)** *Note that $\boldsymbol{\mathcal{D}}_\tau = \big\{\boldsymbol{\theta} \in \mathbb{R}^{d(m+L)} \colon \exists \theta \in \mathcal{C}_\tau \text{ such that } \boldsymbol{\theta} = (0_d, \ldots, 0_d, \theta, \ldots, \theta)\big\}$ is not the only improved confidence set that one can build from $\mathcal{C}_\tau$. Indeed, it is immediate to check that our proof remains unchanged if one uses instead $\boldsymbol{\mathcal{D}}_\tau^{opt} = \big\{\boldsymbol{\theta} \in \mathbb{R}^{d(m+L)} \colon \exists \theta_1, \ldots, \theta_L \in \mathcal{C}_\tau \text{ such that } \boldsymbol{\theta} = (0_d, \ldots, 0_d, \theta_1, \ldots, \theta_L)\big\}$. Optimizing (12) over $\boldsymbol{\mathcal{D}}_{\tau-1}^{opt}$ and not $\boldsymbol{\mathcal{D}}_{\tau-1}$ creates an over-optimistic block version of the UCB, composed of the sum of the UCBs of the single-actions in the block, although the latter might be attained at different models $\theta_i$, while we know that $\theta^*$ is the same model $\theta^*$ repeated $L$ times. Still, since each $\theta_i$ is estimated in the confidence set $\mathcal{C}_{\tau-1}$ of reduced dimension, the guarantees are unchanged. In the rest of the paper, we refer to this variant as the over-optimistic version of OFUL-memory, denoted by* O3M. *Empirically,* O3M *outperforms the vanilla approach. We attribute these better performances to the fact that the confidence set it is built upon is more optimistic.*

**Remark 4 (Generic matrix mapping $A$)** *Note that our analysis naturally extends to any matrix mapping $A$, as long as it is known. The term $(m+1)^{\gamma^+}$ in Theorem 1 is then replaced with $\sup_{a_1 \ldots a_m} \|A(a_1, \ldots, a_m)\|_*$. We highlight however that having access to such knowledge is unlikely in practice. This is why we focus on the simpler parametric family (2), which encompasses many rotting and rising scenarios while allowing us to learn simultaneously $m$ and $\gamma$, as shown in the next section. It is of course possible to extend the family of monomials (2) to a family of polynomials, but this requires tracking more parameters (namely, the different coefficients of the polynomial), thus degrading the final regret bound.*

**Remark 5 (Solving LBM with a general Reinforcement Learning (RL) approach)** *Our setting may be seen as an MDP with a $d$-dimensional continuous space of actions, a $(md)$-dimensional continuous state space (for the past $m$ actions), a deterministic transition function parameterized by an unknown scalar $\gamma$, and a stochastic reward function with a linear dependence on an additional $d$-dimensional latent parameter $\theta^*$. The optimal policy in this MDP is generally nonstationary, and we are not aware of RL algorithms whose regret can be bounded without relying on more specific assumptions on the MDP. By exploiting the structure of the MDP, and restricting to cyclic policies, we show instead that the original problem can be solved using stationary bandit techniques.*

### 3.3 Model Selection

In the absence of prior knowledge on the nature of the nonstationary mechanism at work, a natural idea consists in instantiating several LBMs with different values of $\gamma$ and running a model selection algorithm for bandits (Foster et al., 2019; Cutkosky et al., 2020; Pacchiano et al., 2020). In bandit model selection, where a master algorithm runs the different LBMs, the adaptation to the memory size $m$ becomes more complex. Indeed, the different putative values for $m$ induce different block sizes (see Theorem 1) which perturb the

---

**Algorithm 1** `OFUL-memory` (`OM`, `O3M`)

---

**input** : action space $\mathcal{A} \subset \mathbb{R}^d$, memory size $m$, exponent $\gamma$, regularization parameter $\lambda$, horizon $T$.

**init** : set $L = \sqrt{m/4\,d}\ T^{1/4} - m$, $\ \widehat{\theta}_0 = 0_d$, $\ V_0 = \lambda I_d$, $\ \beta_0 = 0$.

**for** $\tau = 1, \ldots, T/(m+L)$ **do**

> `// OM`                                          `// O3M`
>
> $\boldsymbol{a}_\tau = \arg\max\limits_{a_{\tau,i} \in \mathcal{A}} \sup\limits_{\theta \in \mathcal{C}_{\tau-1}} \sum\limits_{i=1}^{L} \langle a_{\tau,i}, A_{\tau,i-1}\theta \rangle$   or   $\boldsymbol{a}_\tau = \arg\max\limits_{a_{\tau,i} \in \mathcal{A}} \sup\limits_{\theta_i \in \mathcal{C}_{\tau-1}} \sum\limits_{i=1}^{L} \langle a_{\tau,i}, A_{\tau,i-1}\theta_i \rangle$
>
> `// Play and update confidence set`
>
> Play $\boldsymbol{a}_\tau$, collect $y_{\tau,1}, \ldots, y_{\tau,m+L}$
>
> Compute $\mathcal{C}_\tau$, i.e., $\widehat{\theta}_\tau$, $V_\tau$, and $\beta_\tau$ via (10) and (18).

---

time and reward scales of the master algorithm. For instance, bandits with larger block lengths will collect more rewards per block, although they might not be more efficient on average. Our solution consists in feeding the master algorithm with averaged rewards. One may then control the true regret (i.e., not averaged) of the output sequence, against a scaled version of the optimal sequence through Lemma 1, which links the normalized regret of a block meta-algorithm to the true regret of the corresponding sequence of blocks.

**Lemma 1** *Suppose that a block-based bandit algorithm (in our case the bandit combiner) produces a sequence of $T_{\mathrm{bc}}$ blocks $\boldsymbol{a}_\tau$, with possibly different cardinalities $|\boldsymbol{a}_\tau|$, such that*

$$\sum_{\tau=1}^{T_{\mathrm{bc}}} \frac{\widetilde{r}(\widetilde{\boldsymbol{a}})}{|\widetilde{\boldsymbol{a}}|} - \sum_{\tau=1}^{T_{\mathrm{bc}}} \frac{\widetilde{r}(\boldsymbol{a}_\tau)}{|\boldsymbol{a}_\tau|} \leq F(T_{\mathrm{bc}})\,,$$

*for some sublinear function $F$. Then, we have*

$$\frac{\min_\tau |\boldsymbol{a}_\tau|}{\max_\tau |\boldsymbol{a}_\tau|} \left( \widetilde{r}(\widetilde{\boldsymbol{a}})\, \frac{\sum_\tau |\boldsymbol{a}_\tau|}{|\widetilde{\boldsymbol{a}}|} \right) - \sum_{\tau=1}^{T_{\mathrm{bc}}} \widetilde{r}(\boldsymbol{a}_\tau) \ \leq\ \min_\tau |\boldsymbol{a}_\tau|\, F(T_{\mathrm{bc}})\,.$$

*In particular, if all blocks have the same cardinality the last bound is just the block regret bound scaled by $|\boldsymbol{a}_\tau|$.*

Combining this result with Theorem 1 and (Cutkosky et al., 2020, Corollary 2) yields the following result.

**Corollary 1** *Consider an instance of LBM with unknown parameters $(m_\star, \gamma_\star)$. Assume a bandit combiner is run on $N \leq d\sqrt{m_\star}$ instances of OFUL-memory (Algorithm 2), each using a different pair of parameters $(m_i, \gamma_i)$ from a set $\mathcal{S} = \{(m_1, \gamma_1), \ldots, (m_N, \gamma_N)\}$ such that $(m_\star, \gamma_\star) \in \mathcal{S}$. Let $M = (\max_j m_j)/(\min_j m_j)$. Then, for all $T \geq (m_\star + 1)^{2\gamma_\star^+}/m_\star d^4$, the expected rewards $(r_t^{\mathrm{bc}})_{t=1}^{T}$ of the bandit combiner satisfy*

$$\frac{\mathrm{OPT}}{\sqrt{M}} - \mathbb{E}\left[ \sum_{t=1}^{T} r_t^{\mathrm{bc}} \right] \ =\ \widetilde{\mathcal{O}}\!\left( M\, d\, (m_\star + 1)^{1 + \frac{3}{2}\gamma_\star^+}\, T^{3/4} \right)\,.$$

## 4 Algorithms

In this section, we discuss the practical implementation of our approach. This includes OFUL-memory (`OM`) and its over-optimistic variant (`O3M`, see Remark 3), both summarized in Algorithm 1. We also instantiate the `Bandit Combiner` from Cutkosky et al. (2020) to our specific setting with average rewards and `O3M` as base algorithm, see Algorithm 2.

**Maximizing the UCBs.** We start by making explicit the UCBs used in `OM` and `O3M`, see (12), optimized over $\mathcal{D}_\tau$ or $\mathcal{D}_\tau^{\mathrm{opt}}$. Using the formula for $\mathcal{C}_\tau$ one can check that they are given by $\mathrm{UCB}_\tau(\boldsymbol{a}) = \sum_{j=m+1}^{m+L} \langle a_j, A_{j-1}\widehat{\theta}_{\tau-1} \rangle + B(\boldsymbol{a})$, where $B(\boldsymbol{a}) = \beta_{\tau-1} \big\| \sum_{j=m+1}^{m+L} A_{j-1}^\top a_j \big\|_{V_{\tau-1}^{-1}}$ for `OM` and

$B(\boldsymbol{a}) = \beta_{\tau-1} \sum_{j=m+1}^{m+L} \left\| A_{j-1}^\top a_j \right\|_{V_{\tau-1}^{-1}}$ for O3M. The two UCBs only differ in their exploration bonuses. Note that by the triangle inequality, we have $\text{UCB}_\tau^{\text{OM}}(\boldsymbol{a}) \leq \text{UCB}_\tau^{\text{O3M}}(\boldsymbol{a})$ for any $\boldsymbol{a}$. Thanks to this closed form in terms of $\boldsymbol{a}$, it is possible to approximate $\arg\max_{\boldsymbol{a}} \text{UCB}_\tau(\boldsymbol{a})$ using gradient ascent. Note however that maximizing the UCBs is a hard problem when the action space is infinite, which might be non-convex in general. In that respect, the theoretical guarantees we provide in Theorem 1 hold whenever the learner has access to some oracle that returns the exact UCB maximizer, as traditionally assumed in the literature, see e.g., Kveton et al. (2015). Conversely, note that the practical implementation of O3M still satisfies Theorem 1, but for a slightly weaker version of the regret where the "best block" is understood as the one returned by the approximated oracle used in O3M (i.e., our gradient ascent solver). See (Kveton et al., 2015, Section 9) for a similar discussion.

**Computational complexity.** As described in Algorithm 1, our approach consists of two steps: updating the confidence region $C_\tau$, i.e., $\widehat{\theta}_\tau$ and $\beta_\tau$ according to (10) and (18), and computing the block $\boldsymbol{a}_\tau$ that maximizes the UCB index. The first step is performed by online Ridge regression, and has a computational cost of $\mathcal{O}(Ld^2)$. We note here the advantage of our refined algorithm over the naive concatenated approach, whose Ridge regression update has cost $\mathcal{O}(L^2 d^2)$. The maximization of the UCB indices, performed through gradient ascent has time complexity per iteration of $\mathcal{O}((m+L)d^2)$. Hence, the overall complexity of an epoch of Algorithm 1 is $\mathcal{O}((m+L)d^2 \cdot n_{\text{it}})$, where $n_{\text{it}}$ is the number of iterations performed by gradient ascent. Recall that the epochs of Algorithm 1 correspond to blocks of $m+L$ actions, such that the actual per-round complexity is $\mathcal{O}(d^2 \cdot n_{\text{it}})$.

**Bandit combiner.** Our bandit combiner, see Algorithm 2, builds upon the approach developed by Cutkosky et al. (2020) and works as follows. The meta-algorithm is fed with different bandit algorithms (in our case, instances of O3M with different choices of parameters $m_j$ and $\gamma_j$) and at each round plays a block according to one of the algorithms. Each O3M instance comes with a *putative* regret bound $C_j T^{\alpha_j}$, which is the regret bound satisfied by the algorithm *should it be well-specified*, i.e., if the rewards are indeed generated through a memory matrix with memory $m_j$ and exponent $\gamma_j$. Note that in order to be comparable across the different instances, the putative regrets apply to the average rewards. The values of $C_j$ and $\alpha_j$ can be computed using Theorem 1, see the proof of Corollary 1 for details. The putative regrets are then used to successively discard the instances that are not well specified, and eventually identify the instance using parameters $(m_\star, \gamma_\star)$. Knowing $C_j$ and $\alpha_j$, we can compute for any $j$ the target regret

$$R_j = C_j T_{\text{bc}}^{2/3} + \frac{5\sqrt{30}}{18} C_j^{3/2} T_{\text{bc}}^{2/3} + 1152(m_j+1)^{2\gamma_j^+} T^{1/3} \log(T_{\text{bc}}^3 N/\delta) + (N-1)T^{2/3}, \qquad (13)$$

where $T_{\text{bc}}$ is the number of blocks the Bandit Combiner is called on, see Appendix B for details. Here, we note how the presence of $(m_j+1)^{2\gamma_j^+}$ is impacting differently the rising and rotting scenarios. Using (Cutkosky et al., 2020, Corollary 2), the regret of Algorithm 2 is finally given by $3R_{j_\star}$, where $j_\star$ is the index such that $(m_{j_\star}, \gamma_{j_\star}) = (m_\star, \gamma_\star)$.

## 5 Experiments

We perform experiments to validate the theoretical performance of OM and O3M (Algorithm 1). Similarly to (Warlop et al., 2018), we work with synthetic data because of the counterfactual nature of the learning problem in bandits. Unless stated otherwise, we set $d = 3$ while $\theta^* \in \mathbb{R}^d$ is generated uniformly at random with unit norm. The rewards are generated according to (1) and (2), and perturbed by Gaussian noise with standard deviation $\sigma = 1/10$.

**Rotting with Bandit Combiner.** We start by analyzing the rotting scenario with $m = 2$ and $\gamma = -3$. We measure the performance in terms of the cumulative reward averaged over 5 runs (this is enough because the variance is small). In Figure 2 (left pane) we compare the performance of O3M against oracle greedy, vanilla OFUL, and two instances of Bandit Combiner (Algorithm 2. The first instance, Combiner $\gamma$, works in the setting where the misspecified parameter is $\gamma$ and the algorithm is run over the set $\{-4, -3, -2, -1, 0\}$ of possible values for $\gamma$ with the true value being $-3$. The second instance, Combiner $m$, tests the setting where

---

**Algorithm 2** `Bandit Combiner on O3M`

---

**input** : Instances $\texttt{O3M}(m_1, \gamma_1), \ldots, \texttt{O3M}(m_N, \gamma_N)$, horizon $T_{\mathrm{bc}}$
numbers $C_1, \ldots, C_N > 0$, target regrets $R_1, \ldots, R_N$.

Set $T(i) = 0, \mathcal{S}_i = 0, \Delta_i = 0$ for $i = 1, \ldots, N$, and set $I_0 = \{1, \ldots, N\}$

**for** $t = 1, \ldots, T_{bc}$ **do**

    **if** *there is some $i \in I_t$ with $T(i) = 0$* **then**
        $i_t = i$
    **else**
        For each $i \in I_t$, compute the UCB index:

$$\mathrm{UCB}(i) = \min\left\{(m_i + 1)^{2\gamma_i^+}, \frac{C_i}{\sqrt{T(i)}} + 4(m_i + 1)^{2\gamma_i^+}\sqrt{\frac{2\log(T^3 N/\delta)}{T(i)}}\right\} - \frac{R_i}{T_{\mathrm{bc}}}$$

        Set $i_t = \arg\max_{i \in I_t} \frac{\mathcal{S}_i}{T(i)} + \mathrm{UCB}(i)$
    Obtain from instance $\texttt{O3M}(m_{i_t}, \gamma_{i_t})$ a block of size $m_{i_t} + L_{i_t}$ and play it

    Return the total reward $r_{i_t}$ collected in the last $L_{i_t}$ time steps of the block to $\texttt{O3M}(m_{i_t}, \gamma_{i_t})$

    Compute the average reward $\widehat{r}_{i_t} = \frac{r_{i_t}}{L_{i_t}}$

    Update $\Delta_{i_t} \leftarrow \Delta_{i_t} + \mathcal{S}_{i_t}/T(i_t) - \widehat{r}_{i_t}$ (where we set $0/0 = 0$) and $\mathcal{S}_{i_t} \leftarrow \mathcal{S}_{i_t} + \widehat{r}_{i_t}$

    Update the number of plays $T(i_t) \leftarrow T(i_t) + 1$

    **if** $\Delta_{i_t} \geq C_{i_t} T(i_t)^{\gamma_{i_t}} + 12(m_{i_t} + 1)^{2\gamma_{i_t}^+}\sqrt{2\log(T^3 N/\delta)T(i_t)}$ **then**
        $I_t = I_{t-1} \setminus \{i_t\}$
    **else**
        $I_t = I_{t-1}$

---

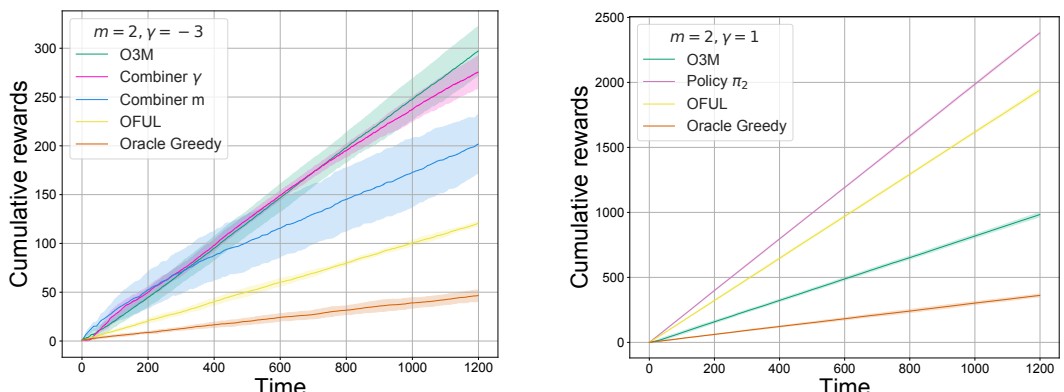

Figure 2: Cumulative rewards in rotting (left) and rising with non-isotropic initialization (right) cases.

the misspecified parameter is $m$. In this case the algorithm is run over the set $\{0, 2, 3\}$ of possible values for $m$ with the true value being 2. The results—see Figure 2 (left pane)—show that $\texttt{O3M}$ is able to plan the actions in the block ensuring that a good arm is not played right away if a higher reward can be obtained later on in the block. This means that $\texttt{O3M}$ is waiting to play certain actions until the corresponding entries of $A$ have been offloaded, preventing $A$ to negatively impact the reward of these actions. Although learning $m$ proves to be more difficult, which is consistent with the impact of $M = (\max_j m_j)/(\min_j m_j)$ in Corollary 1, Combiner $m$ run on instances of $\texttt{O3M}$ is competitive with $\texttt{O3M}$ run with the true parameters. Note that with isotropic initialization there is no point in running Combiner $\gamma$ with values of $\gamma$ larger than zero. Indeed, in the isotropic case oracle greedy is optimal, stationary, and with the same optimal action for any $\gamma \geq 0$. The empirical performance of our algorithms in a non-isotropic rising setting is investigated in the next example.

**Rising with non-isotropic initialization.** When $\gamma > 0$ (rising setting) and $A_0 \neq I_d$ (non-isotropic initialization), there are instances for which oracle greedy is suboptimal, as we show next. Let $d = 2$, $m = 2$, $\gamma = 1$, $A_0 = e_1 e_1^\top$, and $\theta^* = (\sqrt{\epsilon}, \sqrt{1 - \epsilon})$. With these choices, oracle greedy starts to pull action $e_1 = (1, 0)$ and will always play it, obtaining a cumulative reward of $T(1 + m)\sqrt{\epsilon}$. Instead, a better strategy would be to play $e_2 = (0, 1)$ all the time, collecting a cumulative reward of $Tm\sqrt{1 - \epsilon}$. We call this strategy $\pi_2$ and in Figure 2 (right pane) we compare the performance of O3M with oracle greedy, $\pi_2$, and OFUL. Here OFUL performs well because the optimal action is stationary and, unlike oracle greedy, OFUL can use exploration to discover that $e_2$ is better than $e_1$.

## 6 Conclusions and open problems

We introduced and analyzed a nonstationary generalization of linear bandits that uses a fixed-size memory. Some interesting future research directions may include the derivation of a matching lower bound or quantifying the UCB optimization error to better tradeoff the block length $L$.

### Acknowledgments

The authors acknowledge the financial support from the MUR PRIN grant 2022EKNE5K (Learning in Markets and Society), the FAIR (Future Artificial Intelligence Research) project, funded by the NextGenerationEU program within the PNRR-PE-AI scheme, and the EU Horizon CL4-2022-HUMAN-02 research and innovation action under grant agreement 101120237, project ELIAS (European Lighthouse of AI for Sustainability).

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

# A  Technical Proofs

We gather in this section the proofs omitted in the core text.

## A.1  Proof of Proposition 1

**Proposition 1** *The oracle greedy strategy, which plays $a_t^{\text{greedy}} = \arg\max_{a \in \mathcal{A}} \langle a, A_{t-1}\theta^* \rangle$ at time step $t$, can suffer linear regret, both in rotting or rising scenarios.*

**Proof**  We build two instances of LBM, one rotting, one rising, in which the oracle greedy strategy suffers linear regret. We highlight that the other strategy exhibited, which performs better than oracle greedy, may not be optimal.

**Rotting instance.** Let $\mathcal{A} = \mathcal{B}_d$, $\theta^* = e_1$, $m = d - 1$, and $A$ such that

$$A(a_1, \ldots, a_m) = \left( I_d + \sum_{s=1}^{m} a_s a_s^\top \right)^{-\gamma},$$

for some $\gamma > 0$ to be specified later. Oracle greedy, which plays at each time step $a_t^{\text{greedy}} = \arg\max_{a \in \mathcal{A}} \langle a, A_{t-1}\theta^* \rangle$, constantly plays $e_1$. After the first $m$ pulls, it collects a reward of $1/d^\gamma$ at every time step. On the other side, the strategy that plays cyclically the block $e_1 \ldots e_d$ collects a reward of 1 every $d = m + 1$ time steps, i.e., an average reward of $1/d$ per step. Hence, up to the transitive first $m$ puuls, the cumulative reward of oracle greedy after $T$ rounds is $T/d^\gamma$, and that of the cyclic policy is $T/d$. The regret of oracle greedy is thus at least

$$T\left( \frac{1}{d} - \frac{1}{d^\gamma} \right),$$

which is linear for $\gamma > 1$.

**Rising instance.** Let $m \geq 1$, $d = 2$, $\mathcal{A} = \mathcal{B}_2$, $\theta^* = (\varepsilon, 1)$ where $\varepsilon > 0$ is to be specified later, and $A$ such that

$$A(a_1, \ldots, a_m) = \begin{pmatrix} 1 & 0 \\ 0 & 0 \end{pmatrix} + \sum_{s=1}^{m} a_s a_s^\top.$$

Oracle greedy constantly plays $e_1$ collecting a reward of $(m+1)\theta_1^*$ from round $m+1$ onward. On the other side, the strategy that plays constantly $e_2$ collects a reward of $m\theta_2^*$ from round $m+1$ onward. Hence, the regret of oracle greedy from round $m+1$ onward is at least $(T - m)[m - (m+1)\varepsilon]$, which is linear for $\varepsilon < m/(m+1)$.  □

## A.2  Proof of Proposition 2

**Proposition 2** *For any $m, L \geq 1$, let $\widetilde{a}$ be the block of $m + L$ actions defined in (5) and $(\widetilde{r}_t)_{t=1}^T$ be the expected rewards collected when playing cyclically $\widetilde{a}$. We have*

$$\text{OPT} - \sum_{t=1}^{T} \widetilde{r}_t \leq \frac{2mR}{m+L} T. \tag{6}$$

**Proof**  Recall that the optimal sequence is denoted $(a_t^*)_{t=1}^T$ and collects rewards $(r_t^*)_{t=1}^T$. Let $L > 0$; by definition, there exists a block of actions of length $L$ in $(a_t^*)_{t=1}^T$ with average expected reward higher that $\text{OPT}/T$. Let $t^*$ be the first index of this block, we thus have $(1/L)\sum_{t=t^*}^{t^*+L-1} r_t^* \geq \text{OPT}/T$. However, this average expected reward is realized only using the initial matrix $A_{t^*-1}$, generated from $a_{t^*-1}^*, \ldots, a_{t^*-m}^*$. Let $\boldsymbol{a}^* = a_{t^*-m}^*, \ldots, a_{t^*+L-1}^*$ of length $m + L$. Note that, by definition, we have that $\widetilde{r}(\widetilde{\boldsymbol{a}}) \geq \widetilde{r}(\boldsymbol{a}^*) = \sum_{t=t^*}^{t^*+L-1} r_t^* \geq L\,\text{OPT}/T$. Furthermore, by (8), when playing cyclically $\widetilde{a}$ one obtains at least a reward of $-R$ in each one of the first $m$ pulls of the block. Collecting all the pieces, we obtain

$$\sum_{t=1}^{T} \widetilde{r}_t \geq \frac{T}{m+L}\left( -mR + \widetilde{r}(\widetilde{\boldsymbol{a}}) \right)$$

$$\geq \frac{T}{m+L}\Big(-mR+\widetilde{r}(\boldsymbol{a}^*)\Big)$$

$$\geq \frac{T}{m+L}\left(-mR+L\,\frac{\mathrm{OPT}}{T}\right)$$

$$= \frac{L}{m+L}\mathrm{OPT}-\frac{mR}{m+L}\,T$$

$$\geq \frac{L}{m+L}\mathrm{OPT}+\frac{m}{m+L}\mathrm{OPT}-\frac{mR}{m+L}\,T-\frac{mR}{m+L}\,T \qquad (14)$$

$$= \mathrm{OPT}-\frac{2mR}{m+L}\,T\,,$$

where (14) derives from $\mathrm{OPT}\leq RT$. □

### A.3 Proof of Proposition 4

We prove the (stronger) high probability version of Proposition 4.

**Proposition 5** *Let $\lambda\geq 1$, $\delta\in(0,1)$, and $\boldsymbol{a}_\tau$ be the blocks of actions in $\mathbb{R}^{d(m+L)}$ associated to the $\boldsymbol{b}_\tau$ defined in* (9)*. Then, with probability at least $1-\delta$ we have*

$$\sum_{\tau=1}^{T/(m+L)} \widetilde{r}(\widetilde{\boldsymbol{a}})-\widetilde{r}(\boldsymbol{a}_\tau) \leq 4L(m+1)^{\gamma^+}\sqrt{Td\,\ln\left(1+\frac{T(m+1)^{2\gamma^+}}{d(m+L)\lambda}\right)}$$

$$\cdot\left(\sqrt{\lambda L}+\sqrt{\ln\left(\frac{1}{\delta}\right)+d(m+L)\ln\left(1+\frac{T(m+1)^{2\gamma^+}}{d(m+L)\lambda}\right)}\right)\,.$$

**Proof** The proof essentially follows that of (Abbasi-Yadkori et al., 2011, Theorem 3). The main difference is that our version of OFUL operates at the block level. This implies a smaller time horizon, but also and increased dimension and an instantaneous regret $\langle\widetilde{\boldsymbol{b}},\boldsymbol{\theta}^*\rangle-\langle\boldsymbol{b}_\tau,\boldsymbol{\theta}^*\rangle$ upper bounded by $2L(m+1)^{\gamma^+}$ instead of 1. We detail the main steps of the proof for completeness. Recall that running OFUL in our case amounts to compute at every block time step $\tau$

$$\widehat{\boldsymbol{\theta}}_\tau = \boldsymbol{V}_\tau^{-1}\left(\sum_{\tau'=1}^{\tau}\boldsymbol{y}_{\tau'}\,\boldsymbol{b}_{\tau'}\right),$$

where

$$\boldsymbol{V}_\tau = \sum_{\tau'=1}^{\tau}\boldsymbol{b}_{\tau'}\boldsymbol{b}_{\tau'}^\top+\lambda I_{d(m+L)}\,,\qquad \text{and}\qquad \boldsymbol{y}_\tau = \sum_{i=m+1}^{m+L}y_{\tau,i}\,,$$

since we associate with a block of actions the sum of rewards obtained after time step $m$. Note that by the determinant-trace inequality, see e.g., (Abbasi-Yadkori et al., 2011, Lemma 10), with actions $\boldsymbol{b}_\tau$ that satisfy $\|\boldsymbol{b}_\tau\|_2^2\leq m+L(m+1)^{2\gamma^+}$ we have

$$\frac{|\boldsymbol{V}_\tau|}{|\lambda I_{d(m+L)}|} \leq \left(1+\frac{\tau(m+L(m+1)^{2\gamma^+})}{d(m+L)\lambda}\right)^{d(m+L)} \leq \left(1+\frac{\tau(m+1)^{2\gamma^+}}{d\lambda}\right)^{d(m+L)}. \qquad (15)$$

The action played at block time step $\tau$ is the block $\boldsymbol{a}_\tau\in\mathcal{B}_d^{m+L}$ associated with

$$\boldsymbol{b}_\tau = \arg\max_{\boldsymbol{b}\in\mathcal{B}}\,\sup_{\boldsymbol{\theta}\in\mathcal{C}_{\tau-1}}\,\langle\boldsymbol{b},\boldsymbol{\theta}\rangle\,, \qquad (16)$$

where

$$\mathcal{C}_\tau = \left\{\boldsymbol{\theta}\in\mathbb{R}^{d(m+L)}\colon \left\|\widehat{\boldsymbol{\theta}}_\tau-\boldsymbol{\theta}\right\|_{\boldsymbol{V}_\tau}\leq\beta_\tau(\delta)\right\}\,,$$

with

$$\beta_\tau(\delta) = \sqrt{2\ln\left(\frac{1}{\delta}\right) + d(m+L)\ln\left(1 + \frac{\tau(m+1)^{2\gamma^+}}{d\lambda}\right)} + \sqrt{\lambda L}\,. \tag{17}$$

Applying (Abbasi-Yadkori et al., 2011, Theorem 2) to $\boldsymbol{\theta}^* \in \mathbb{R}^{d(m+L)}$ which satisfies $\|\boldsymbol{\theta}^*\|_2 \leq \sqrt{L}$ we have that $\boldsymbol{\theta}^* \in \mathcal{C}_\tau$ for every $\tau$ with probability at least $1 - \delta$. Denoting by $\widetilde{\boldsymbol{\theta}}_\tau$ the model that maximizes (16), we thus have that with probability at least $1 - \delta$, the inequality $\langle \widetilde{\boldsymbol{b}}, \boldsymbol{\theta}^* \rangle \leq \langle \boldsymbol{b}_\tau, \widetilde{\boldsymbol{\theta}}_\tau \rangle$ holds for every $\tau$, and consequently

$$\sum_{\tau=1}^{T/(m+L)} \langle \widetilde{\boldsymbol{b}}, \boldsymbol{\theta}^* \rangle - \langle \boldsymbol{b}_\tau, \boldsymbol{\theta}^* \rangle$$

$$\leq \sum_{\tau=1}^{T/(m+L)} \min\left\{2L(m+1)^{\gamma^+}, \langle \boldsymbol{b}_\tau, \widetilde{\boldsymbol{\theta}}_\tau - \boldsymbol{\theta}^* \rangle\right\}$$

$$\leq \sum_{\tau=1}^{T/(m+L)} \min\left\{2L(m+1)^{\gamma^+}, \left\|\widetilde{\boldsymbol{\theta}}_\tau - \boldsymbol{\theta}^*\right\|_{\boldsymbol{V}_{\tau-1}} \|\boldsymbol{b}_\tau\|_{\boldsymbol{V}_{\tau-1}^{-1}}\right\}$$

$$\leq \sum_{\tau=1}^{T/(m+L)} \min\left\{2L(m+1)^{\gamma^+}, 2\beta_\tau(\delta) \|\boldsymbol{b}_\tau\|_{\boldsymbol{V}_{\tau-1}^{-1}}\right\}$$

$$\leq 2L(m+1)^{\gamma^+} \beta_{T/(m+L)}(\delta) \sum_{\tau=1}^{T/(m+L)} \min\left\{1, \|\boldsymbol{b}_\tau\|_{\boldsymbol{V}_{\tau-1}^{-1}}\right\}$$

$$\leq 2L(m+1)^{\gamma^+} \beta_{T/(m+L)}(\delta) \sqrt{\frac{T}{m+L} \sum_{\tau=1}^{T/(m+L)} \min\left\{1, \|\boldsymbol{b}_\tau\|_{\boldsymbol{V}_{\tau-1}^{-1}}^2\right\}}$$

$$\leq 2\sqrt{2}L(m+1)^{\gamma^+} \beta_{T/(m+L)}(\delta) \sqrt{\frac{T}{m+L} \ln\frac{|\boldsymbol{V}_{T/(m+L)}|}{|\lambda I_{d(m+L)}|}}$$

$$\leq 4L(m+1)^{\gamma^+} \sqrt{Td \ln\left(1 + \frac{T(m+1)^{2\gamma^+}}{d(m+L)\lambda}\right)}$$

$$\cdot \left(\sqrt{\lambda L} + \sqrt{\ln\left(\frac{1}{\delta}\right) + d(m+L)\ln\left(1 + \frac{T(m+1)^{2\gamma^+}}{d(m+L)\lambda}\right)}\right),$$

where we have used (Abbasi-Yadkori et al., 2011, Lemma 11), as well as (15) and (17). Note that in the stationary case, i.e., when $m = 0$ and $L = 1$, we exactly recover (Abbasi-Yadkori et al., 2011, Theorem 3). Proposition 4 is obtained by setting $\lambda \in [1, d]$, $L \geq m$, and $\delta = 1/T$. $\qquad\square$

## A.4   Proof of Proposition 3

**Proof**   Let $d = m + 1$, $\mathcal{A} = \{0_d\} \cup (e_k)_{k \leq d}$, $\theta^* = (1/\sqrt{d}, \ldots, 1/\sqrt{d})$, and $\gamma \leq 0$. For simplicity, we note the basis modulo $d$, i.e., $e_{k+d} = e_k$ for any $k \in \mathbb{N}$. Note that for any $a_1, \ldots, a_{m+1} \in \mathcal{A}$ we have $\left|\langle a_{m+1}, A_m \theta^* \rangle\right| \leq \|a_{m+1}\|_1 \|A_m \theta^*\|_\infty \leq 1/\sqrt{d}$, such that one can take $R = 1/\sqrt{d}$. Observe now that the strategy which plays cyclically $e_1, \ldots, e_d$ collects a reward of $1/\sqrt{d}$ at each time step, which is optimal, such that OPT $= T/\sqrt{d}$. Further, it is easy to check that block $\widetilde{a}$, composed of $m$ pulls of $0_d$ followed by $e_1, \ldots, e_L$ satisfies $\widetilde{r}(\widetilde{a}) = L/\sqrt{d}$, which is optimal for similar reasons. Playing cyclically $\widetilde{a}$, one gets a reward of $L/\sqrt{d}$ every $m + L$ pulls. In other terms, we have

$$\text{OPT} - \sum_{t=1}^{T} \widetilde{r}_t = \frac{T}{\sqrt{d}} - \frac{L}{m+L}\frac{T}{\sqrt{d}} = \frac{m}{m+L}\frac{T}{\sqrt{d}} = \frac{mR}{m+L}T\,.$$

$\qquad\square$

### A.5 Proof of Theorem 1

We prove the high probability version of Theorem 1, obtained by setting $\lambda \in [1, d]$, and $\delta = 1/T$.

**Theorem 2** *Let $\lambda \geq 1$, $\delta \in (0, 1)$, and $\boldsymbol{a}_\tau$ be the blocks of actions in $\mathbb{R}^{d(m+L)}$ defined in (11). Then, with probability at least $1 - \delta$ we have*

$$\sum_{\tau=1}^{T/(m+L)} \widetilde{r}(\widetilde{\boldsymbol{a}}) - \widetilde{r}(\boldsymbol{a}_\tau) \leq 4L(m+1)^{\gamma^+} \sqrt{Td \, \ln\left(1 + \frac{T(m+1)^{2\gamma^+}}{d\lambda}\right)} \\ \cdot \left(\sqrt{\lambda} + \sqrt{\ln\left(\frac{1}{\delta}\right) + d \, \ln\left(1 + \frac{T(m+1)^{2\gamma^+}}{d(m+L)\lambda}\right)}\right).$$

*Let $m \geq 1$, $T \geq m^2 d^2 + 1$, and set $L = \left\lceil \sqrt{m/d} \, T^{1/4} \right\rceil - m$. Let $r_t$ be the rewards collected when playing $\boldsymbol{a}_\tau$ as defined in (11). Then, with probability at least $1 - \delta$ we have*

$$\text{OPT} - \sum_{t=1}^{T} r_t \leq 4\sqrt{d} \, (m+1)^{\frac{1}{2}+\gamma^+} \, T^{3/4} \left[1 + 2\sqrt{\ln\left(1 + \frac{T(m+1)^{2\gamma^+}}{d\lambda}\right)} \right. \\ \left. \cdot \left(\sqrt{\frac{\lambda}{d}} + \sqrt{\frac{\ln(1/\delta)}{d} + \ln\left(1 + \frac{T(m+1)^{2\gamma^+}}{d\lambda}\right)}\right)\right].$$

**Proof** The proof is along the lines of OFUL's analysis. The main difficulty is that we cannot use the elliptical potential lemma, see e.g., (Lattimore & Szepesvári, 2020, Lemma 19.4) due to the delay accumulated by $V_\tau$, which is computed every $m + L$ round only. Let

$$\beta_\tau(\delta) = \sqrt{2 \ln\left(\frac{1}{\delta}\right) + d \, \ln\left(1 + \frac{\tau(m+1)^{2\gamma^+}}{d\lambda}\right)} + \sqrt{\lambda}. \tag{18}$$

By (Abbasi-Yadkori et al., 2011, Theorem 2), we have with probability at least $1 - \delta$ that $\theta^* \in \mathcal{C}_\tau$ for every $\tau$. It follows directly that $\boldsymbol{\theta}^* \in \boldsymbol{\mathcal{D}}_\tau$ for any $\tau$, such that $\langle \widetilde{\boldsymbol{b}}, \boldsymbol{\theta}^* \rangle \leq \langle \boldsymbol{b}_\tau, \widetilde{\boldsymbol{\theta}}_\tau \rangle$, where $\widetilde{\boldsymbol{\theta}}_\tau = (0_d, \ldots, 0_d, \widetilde{\theta}_\tau, \ldots, \widetilde{\theta}_\tau)$ with $\widetilde{\theta}_\tau \in \mathbb{R}^d$ that maximizes (11) over $\mathcal{C}_{\tau-1}$. It can be shown that the regret is upper bounded by $\sum_\tau \sum_{i=m+1}^{m+L} \langle b_{\tau,i}, \widetilde{\theta}_\tau - \theta^* \rangle$. Following the standard analysis, one could then use

$$\langle b_{\tau,i}, \widetilde{\theta}_\tau - \theta^* \rangle \leq \|b_{\tau,i}\|_{V_{\tau-1}^{-1}} \left\|\widetilde{\theta}_t - \theta^*\right\|_{V_{\tau-1}}.$$

While the confidence set gives $\left\|\widetilde{\theta}_t - \theta^*\right\|_{V_{\tau-1}} \leq 2\beta_{\tau-1}(\delta)$, the quantity $\sum_{i=m+1}^{m+L} \|b_{\tau,i}\|_{V_{\tau-1}^{-1}}$ is much more complex to bound. Indeed, the elliptical potential lemma allows to bound $\sum_t \|a_t\|_{V_{t-1}^{-1}}^2$ when $V_t = \sum_{s \leq t} a_s a_s^\top + \lambda I_d$. However, recall that in our case we have $V_\tau = \sum_{\tau'=1}^{\tau} \sum_{i=m+1}^{m+L} b_{\tau',i} b_{\tau',i}^\top + \lambda I_d$, which is only computed every $m + L$ rounds. As a consequence, there exists a "delay" between $V_{\tau-1}$ and the action $b_{\tau,i}$ for $i \geq m+2$, preventing from using the lemma. Therefore, we propose to use instead

$$\langle b_{\tau,i}, \widetilde{\theta}_\tau - \theta^* \rangle \leq \|b_{\tau,i}\|_{V_{\tau,i-1}^{-1}} \left\|\widetilde{\theta}_t - \theta^*\right\|_{V_{\tau,i-1}}, \quad \text{where} \quad V_{\tau,i} = V_{\tau-1} + \sum_{j=m+1}^{i} b_{\tau,j} b_{\tau,j}^\top. \tag{19}$$

By doing so, the elliptical potential lemma applies. On the other hand, one has to control $\left\|\widetilde{\theta}_t - \theta^*\right\|_{V_{\tau,i-1}}$, which is not anymore bounded by $2\beta_{\tau-1}(\delta)$ since the subscript matrix is $V_{\tau,i-1}$ instead of $V_{\tau-1}$. Still, one can show that for any $i \leq m + L$ we have

$$\left\|\widetilde{\theta}_t - \theta^*\right\|_{V_{\tau,i-1}}^2 \\ = \text{Tr}\left(V_{\tau,i-1} \left(\widetilde{\theta}_t - \theta^*\right)\left(\widetilde{\theta}_t - \theta^*\right)^\top\right)$$

$$
= \mathrm{Tr}\left( \left( V_{\tau-1} + \sum_{j=m+1}^{i-1} b_{\tau,j} b_{\tau,j}^\top \right) \left( \widetilde{\theta}_t - \theta^* \right) \left( \widetilde{\theta}_t - \theta^* \right)^\top \right)
$$

$$
= \mathrm{Tr}\left( \left( I_d + \sum_{j=m+1}^{i-1} \left( V_{\tau-1}^{-1/2} b_{\tau,j} \right) \left( V_{\tau-1}^{-1/2} b_{\tau,j} \right)^\top \right) V_{\tau-1}^{1/2} \left( \widetilde{\theta}_t - \theta^* \right) \left( \widetilde{\theta}_t - \theta^* \right)^\top V_{\tau-1}^{1/2} \right)
$$

$$
\leq \left\| I_d + \sum_{j=m+1}^{i-1} \left( V_{\tau-1}^{-1/2} b_{\tau,j} \right) \left( V_{\tau-1}^{-1/2} b_{\tau,j} \right)^\top \right\|_* \mathrm{Tr}\left( V_{\tau-1}^{1/2} \left( \widetilde{\theta}_t - \theta^* \right) \left( \widetilde{\theta}_t - \theta^* \right)^\top V_{\tau-1}^{1/2} \right)
$$

$$
\leq \left( 1 + \sum_{j=m+1}^{i-1} \left\| V_{\tau-1}^{-1/2} b_{\tau,j} \right\|_2^2 \right) \left\| \widetilde{\theta}_t - \theta^* \right\|_{V_{\tau-1}}^2
$$

$$
\leq \left( 1 + (L-1)(m+1)^{2\gamma^+} \right) \left\| \widetilde{\theta}_t - \theta^* \right\|_{V_{\tau-1}}^2
$$

$$
\leq L(m+1)^{2\gamma^+} \left\| \widetilde{\theta}_t - \theta^* \right\|_{V_{\tau-1}}^2 . \tag{20}
$$

Recalling also that $\langle \widetilde{\boldsymbol{b}}, \boldsymbol{\theta}^* \rangle - \langle \boldsymbol{b}_\tau, \boldsymbol{\theta}^* \rangle \leq 2L(m+1)^{\gamma^+}$, we have with probability at least $1 - \delta$

$$
\sum_{\tau=1}^{T/(m+L)} \langle \widetilde{\boldsymbol{b}}, \boldsymbol{\theta}^* \rangle - \langle \boldsymbol{b}_\tau, \boldsymbol{\theta}^* \rangle
$$

$$
\leq \sum_{\tau=1}^{T/(m+L)} \min\left\{ 2L(m+1)^{\gamma^+}, \langle \boldsymbol{b}_\tau, \widetilde{\boldsymbol{\theta}}_\tau - \boldsymbol{\theta}^* \rangle \right\}
$$

$$
= \sum_{\tau=1}^{T/(m+L)} \min\left\{ 2L(m+1)^{\gamma^+}, \sum_{i=m+1}^{m+L} \langle b_{\tau,i}, \widetilde{\theta}_\tau - \theta^* \rangle \right\}
$$

$$
\leq \sum_{\tau=1}^{T/(m+L)} \min\left\{ 2L(m+1)^{\gamma^+}, \sum_{i=m+1}^{m+L} \left\| b_{\tau,i} \right\|_{V_{\tau,i-1}^{-1}} \left\| \widetilde{\theta}_t - \theta^* \right\|_{V_{\tau,i-1}} \right\}
$$

$$
\leq \sum_{\tau=1}^{T/(m+L)} \min\left\{ 2L(m+1)^{\gamma^+}, 2\sqrt{L}(m+1)^{\gamma^+} \beta_{\tau-1}(\delta) \sum_{i=m+1}^{m+L} \left\| b_{\tau,i} \right\|_{V_{\tau,i-1}^{-1}} \right\}
$$

$$
\leq 2L(m+1)^{\gamma^+} \beta_{T/(m+L)}(\delta) \sum_{\tau=1}^{T/(m+L)} \sum_{i=m+1}^{m+L} \min\left\{ 1, \left\| b_{\tau,i} \right\|_{V_{\tau,i-1}^{-1}} \right\}
$$

$$
\leq 2L(m+1)^{\gamma^+} \beta_{T/(m+L)}(\delta) \sqrt{ \frac{TL}{m+L} \sum_{\tau=1}^{T/(m+L)} \sum_{i=m+1}^{m+L} \min\left\{ 1, \left\| b_{\tau,i} \right\|_{V_{\tau,i-1}^{-1}}^2 \right\} }
$$

$$
\leq 2\sqrt{2}L(m+1)^{\gamma^+} \beta_{T/(m+L)}(\delta) \sqrt{ T \ln \frac{|V_{T/(m+L)}|}{|\lambda I_d|} }
$$

$$
\leq 4L(m+1)^{\gamma^+} \sqrt{ Td \ln\left( 1 + \frac{T(m+1)^{2\gamma^+}}{d\lambda} \right) }
$$

$$
\cdot \left( \sqrt{\lambda} + \sqrt{ \ln\left( \frac{1}{\delta} \right) + d \ln\left( 1 + \frac{T(m+1)^{2\gamma^+}}{d(m+L)\lambda} \right) } \right), \tag{21}
$$

where we have used (18), (19), and (20). Similarly to Proposition 5, note that in the stationary case, i.e., when $m = 0$ and $L = 1$, we exactly recover (Abbasi-Yadkori et al., 2011, Theorem 3). The first claim of Theorem 1 is obtained by setting $\lambda \in [1, d]$, and $\delta = 1/T$.

Let $R_T$ denote the right-hand side of (21). Combining this bound with the arguments of Proposition 2, we have with probability $1 - \delta$

$$\sum_{t=1}^{T} r_t \geq \sum_{\tau=1}^{T/(m+L)} \widetilde{r}(\boldsymbol{a}_\tau) - \frac{m(m+1)^{\gamma^+}}{m+L}\, T \tag{22}$$

$$= \sum_{\tau=1}^{T/(m+L)} \langle \boldsymbol{b}_\tau, \boldsymbol{\theta}^* \rangle - \frac{m(m+1)^{\gamma^+}}{m+L}\, T$$

$$\geq \sum_{\tau=1}^{T/(m+L)} \langle \widetilde{\boldsymbol{b}}, \boldsymbol{\theta}^* \rangle - R_T - \frac{m(m+1)^{\gamma^+}}{m+L}\, T \tag{23}$$

$$= \sum_{\tau=1}^{T/(m+L)} \widetilde{r}(\widetilde{\boldsymbol{a}}) - R_T - \frac{m(m+1)^{\gamma^+}}{m+L}\, T$$

$$\geq \sum_{t=1}^{T} \widetilde{r}_t - R_T - \frac{2m(m+1)^{\gamma^+}}{m+L}\, T \tag{24}$$

$$\geq \text{OPT} - R_T - \frac{4m(m+1)^{\gamma^+}}{m+L}\, T \tag{25}$$

$$\geq \text{OPT} - 4(m+1)^{\gamma^+} \left[ \frac{mT}{m+L} + (m+L)\sqrt{Td\, \ln\left(1 + \frac{T(m+1)^{2\gamma^+}}{d\lambda}\right)} \right.$$

$$\left. \cdot \left(\sqrt{\lambda} + \sqrt{\ln\left(\frac{1}{\delta}\right) + d\ln\left(1 + \frac{T(m+1)^{2\gamma^+}}{d(m+L)\lambda}\right)}\right) \right],$$

where (22) and (24) come from the fact that any instantaneous reward is bounded by $(m+1)^{\gamma^+}$, see (8), (23) from (21), and (25) from Proposition 2.

Now, assume that $m \geq 1$, $T \geq d^2 m^2 + 1$, and let $L = \lceil \sqrt{m/d}\, T^{1/4} \rceil - m$. By the condition on $T$, we have $\sqrt{m/d}\, T^{1/4} > m \geq 1$, such that $L \geq 1$ and

$$\sqrt{\frac{m}{d}}\, T^{1/4} \leq \left\lceil \sqrt{\frac{m}{d}}\, T^{1/4} \right\rceil = L + m \leq \sqrt{\frac{m}{d}}\, T^{1/4} + 1 \leq 2\sqrt{\frac{m}{d}}\, T^{1/4}\,.$$

Substituting in the above bound, we have with probability $1 - \delta$

$$\text{OPT} - \sum_{t=1}^{T} r_t \leq 4\sqrt{d}\,(m+1)^{\frac{1}{2}+\gamma^+}\, T^{3/4} \left[ 1 + 2\sqrt{\ln\left(1 + \frac{T(m+1)^{2\gamma^+}}{d\lambda}\right)} \right.$$

$$\left. \cdot \left(\sqrt{\frac{\lambda}{d}} + \sqrt{\frac{\ln(1/\delta)}{d} + \ln\left(1 + \frac{T(m+1)^{2\gamma^+}}{d\lambda}\right)}\right) \right].$$

The second claim of Theorem 1 is obtained by setting $\lambda \in [1, d]$, and $\delta = 1/T$. $\qquad\square$

### A.6   Proof of Corollary 1

**Lemma 1** *Suppose that a block-based bandit algorithm (in our case the bandit combiner) produces a sequence of $T_{\mathrm{bc}}$ blocks $\boldsymbol{a}_\tau$, with possibly different cardinalities $|\boldsymbol{a}_\tau|$, such that*

$$\sum_{\tau=1}^{T_{\mathrm{bc}}} \frac{\widetilde{r}(\widetilde{\boldsymbol{a}})}{|\widetilde{\boldsymbol{a}}|} - \sum_{\tau=1}^{T_{\mathrm{bc}}} \frac{\widetilde{r}(\boldsymbol{a}_\tau)}{|\boldsymbol{a}_\tau|} \leq F(T_{\mathrm{bc}})\,,$$

*for some sublinear function $F$. Then, we have*

$$\frac{\min_\tau |\boldsymbol{a}_\tau|}{\max_\tau |\boldsymbol{a}_\tau|} \left( \widetilde{r}(\widetilde{\boldsymbol{a}})\, \frac{\sum_\tau |\boldsymbol{a}_\tau|}{|\widetilde{\boldsymbol{a}}|} \right) - \sum_{\tau=1}^{T_{\mathrm{bc}}} \widetilde{r}(\boldsymbol{a}_\tau) \leq \min_\tau |\boldsymbol{a}_\tau|\, F(T_{\mathrm{bc}})\,.$$

*In particular, if all blocks have the same cardinality the last bound is just the block regret bound scaled by $|\boldsymbol{a}_\tau|$.*

**Proof** We have

$$
\begin{aligned}
\sum_{\tau=1}^{T_{\mathrm{bc}}} \widetilde{r}(\boldsymbol{a}_\tau) &\geq \min_\tau |\boldsymbol{a}_\tau| \sum_{\tau=1}^{T_{\mathrm{bc}}} \frac{\widetilde{r}(\boldsymbol{a}_\tau)}{|\boldsymbol{a}_\tau|} \\
&\geq \min_\tau |\boldsymbol{a}_\tau| \left( \sum_{\tau=1}^{T_{\mathrm{bc}}} \frac{\widetilde{r}(\widetilde{\boldsymbol{a}})}{|\widetilde{\boldsymbol{a}}|} - F(T_{\mathrm{bc}}) \right) \\
&= \frac{\min_\tau |\boldsymbol{a}_\tau|}{\max_\tau |\boldsymbol{a}_\tau|} \frac{\widetilde{r}(\widetilde{\boldsymbol{a}})}{|\widetilde{\boldsymbol{a}}|} \max_\tau |\boldsymbol{a}_\tau| \, T_{\mathrm{bc}} - \min_\tau |\boldsymbol{a}_\tau| \, F(T_{\mathrm{bc}}) \\
&\geq \frac{\min_\tau |\boldsymbol{a}_\tau|}{\max_\tau |\boldsymbol{a}_\tau|} \left( \widetilde{r}(\widetilde{\boldsymbol{a}}) \frac{\sum_\tau |\boldsymbol{a}_\tau|}{|\widetilde{\boldsymbol{a}}|} \right) - \min_\tau |\boldsymbol{a}_\tau| \, F(T_{\mathrm{bc}}) \,.
\end{aligned}
$$

$\square$

**Corollary 1** *Consider an instance of LBM with unknown parameters $(m_\star, \gamma_\star)$. Assume a bandit combiner is run on $N \leq d\sqrt{m_\star}$ instances of OFUL-memory (Algorithm 2), each using a different pair of parameters $(m_i, \gamma_i)$ from a set $\mathcal{S} = \{(m_1, \gamma_1), \ldots, (m_N, \gamma_N)\}$ such that $(m_\star, \gamma_\star) \in \mathcal{S}$. Let $M = (\max_j m_j)/(\min_j m_j)$. Then, for all $T \geq (m_\star + 1)^{2\gamma_\star^+}/m_\star d^4$, the expected rewards $\left(r_t^{\mathrm{bc}}\right)_{t=1}^T$ of the bandit combiner satisfy*

$$
\frac{\mathrm{OPT}}{\sqrt{M}} - \mathbb{E}\left[ \sum_{t=1}^T r_t^{\mathrm{bc}} \right] = \widetilde{\mathcal{O}}\!\left( M \, d \, (m_\star + 1)^{1 + \frac{3}{2}\gamma_\star^+} T^{3/4} \right).
$$

**Proof** Let $m_\star$ be the true memory size, and $L_\star = L(m_\star)$ the corresponding (partial) block length. Throughout the proof, $\widetilde{\boldsymbol{a}}$ denotes the block defined in (5) with length $m_\star + L_\star$. First observe that only one of the OFUL-memory instances we test is well-specified, i.e., has the true parameters $(m_\star, \gamma_\star)$. We can thus rewrite the regret bound for the Bandit Combiner (Cutkosky et al., 2020, Corollary 2), generalized to rewards bounded in $[-R, R]$ as follows

$$
\mathrm{Regret}_{\mathrm{bc}} = \widetilde{\mathcal{O}}\left( C_\star T_{\mathrm{bc}}^{\alpha_\star} + C_\star^{\frac{1}{\alpha_\star}} T_{\mathrm{bc}} \eta_\star^{\frac{1-\alpha_\star}{\alpha_\star}} + R^2 T_{\mathrm{bc}} \eta_\star + \sum_{j \neq \star} \frac{1}{\eta_j} \right), \tag{26}
$$

where $T_{\mathrm{bc}} = T/(m_\star + L_\star)$ is the bandit combiner horizon, $C_\star$ and $\alpha_\star$ are the constants in the regret bound of the well-specified instance (see below how we determine them), and the $\eta_j$ are free parameters to be tuned. We now derive $C_\star$ and $\alpha_\star$. To that end, we must establish the regret bound of the well-specified instance, and identify $C_\star$ and $\alpha_\star$ such that this bound is equal to $C_\star T_{\mathrm{bc}}^{\alpha_\star}$, where $C_\star$ may contain logarithmic factors. For the well-specified instance, the first claim of Theorem 2 gives that, with probability at least $1 - \delta$, we have

$$
\begin{aligned}
\sum_{\tau=1}^{T/(m_\star + L_\star)} \widetilde{r}(\widetilde{\boldsymbol{a}}) - \widetilde{r}(\boldsymbol{a}_\tau) &\leq 4(m_\star + L_\star)(m_\star + 1)^{\gamma_\star^+} \sqrt{Td \, \ln\!\left( 1 + \frac{T(m_\star + 1)^{2\gamma_\star^+}}{d\lambda} \right)} \\
&\qquad \left( \sqrt{\lambda} + \sqrt{\ln\!\left( \frac{1}{\delta} \right) + d \ln\!\left( 1 + \frac{T(m_\star + 1)^{2\gamma_\star^+}}{d(m_\star + L_\star)\lambda} \right)} \right)
\end{aligned}
$$

$$
\begin{aligned}
\sum_{\tau=1}^{T/(m_\star + L_\star)} \frac{\widetilde{r}(\widetilde{\boldsymbol{a}})}{|\widetilde{\boldsymbol{a}}|} - \frac{\widetilde{r}(\boldsymbol{a}_\tau)}{|\boldsymbol{a}_\tau|} &\leq T^{1/2} 4(m_\star + 1)^{\gamma_\star^+} \sqrt{d \ln\!\left( 1 + \frac{T(m_\star + 1)^{2\gamma_\star^+}}{d\lambda} \right)} \\
&\qquad \left( \sqrt{\lambda} + \sqrt{\ln\!\left( \frac{1}{\delta} \right) + d \ln\!\left( 1 + \frac{T(m_\star + 1)^{2\gamma_\star^+}}{d(m_\star + L_\star)\lambda} \right)} \right),
\end{aligned} \tag{27}
$$

where we have used that $|\boldsymbol{a}_\tau| = |\widetilde{\boldsymbol{a}}| = m_\star + L_\star$ for every $\tau$. Note that the right-hand side of (27) is expressed in terms of $T$, which is not the correct horizon, $T/(m_\star + L_\star)$. However, recall that we have

$$m_\star + L_\star \le 2\sqrt{\frac{m_\star}{d}}\, T^{1/4}$$

$$(m_\star + L_\star)^4 \le \left(\frac{4m_\star}{d}\right)^2 T$$

$$T^3 \le \left(\frac{4m_\star}{d}\right)^2 \left(\frac{T}{m_\star + L_\star}\right)^4$$

$$T^{1/2} \le \left(\frac{4m_\star}{d}\right)^{1/3} \left(\frac{T}{m_\star + L_\star}\right)^{2/3},$$

such that by substituting in (27) and identifying we have $\alpha_\star = 2/3$, and

$$C_\star = 4\left(\frac{4m_\star}{d}\right)^{1/3} (m_\star + 1)^{\gamma_\star^+} \sqrt{d \ln\left(1 + \frac{T_{\mathrm{bc}}(m_\star + L_\star)(m_\star + 1)^{2\gamma_\star^+}}{d\lambda}\right)}$$

$$\left(\sqrt{\lambda} + \sqrt{\ln\left(\frac{1}{\delta}\right) + d\ln\left(1 + \frac{T_{\mathrm{bc}}(m_\star + 1)^{2\gamma_\star^+}}{d\lambda}\right)}\right).$$

Setting $\eta_j = T_{\mathrm{bc}}^{-2/3}$, and substituting in (26) with $R = (m_\star + 1)^{\gamma_\star^+}$, we have that with high probability

$$\sum_{\tau=1}^{T_{\mathrm{bc}}} \frac{\widetilde{r}(\widetilde{\boldsymbol{a}})}{|\widetilde{\boldsymbol{a}}|} - \frac{\widetilde{r}(\boldsymbol{a}_\tau^{\mathrm{bc}})}{|\boldsymbol{a}_\tau^{\mathrm{bc}}|} = \widetilde{\mathcal{O}}\left((C_\star^{3/2} + N)\, T_{\mathrm{bc}}^{2/3} + (m_\star + 1)^{2\gamma_\star^+}\, T_{\mathrm{bc}}^{1/3}\right).$$

Now, recall that $T_{\mathrm{bc}} = \mathcal{O}\left(\sqrt{d/m_\star}\, T^{3/4}\right)$, and that $C_\star = \widetilde{\mathcal{O}}\left((m_\star + 1)^{\frac{1}{3}+\gamma_\star^+} d^{2/3}\right)$. Hence, $N \le d\sqrt{m_\star}$ implies $N = \mathcal{O}\left(C_j^{3/2}\right)$, and $(m_\star + 1)^{\gamma_\star^+} \le d^2\sqrt{m_\star T}$ implies $(m_\star + 1)^{\gamma_\star^+} T_{\mathrm{bc}}^{1/3} = \mathcal{O}\left(C_\star^{3/2} T_{\mathrm{bc}}^{2/3}\right)$. Setting $\lambda \in [1, d]$, $\delta = 1/T$, we obtain

$$\mathbb{E}\left[\sum_{\tau=1}^{T_{\mathrm{bc}}} \frac{\widetilde{r}(\widetilde{\boldsymbol{a}})}{|\widetilde{\boldsymbol{a}}|} - \frac{\widetilde{r}(\boldsymbol{a}_\tau^{\mathrm{bc}})}{|\boldsymbol{a}_\tau^{\mathrm{bc}}|}\right] = \widetilde{\mathcal{O}}\left(d\sqrt{m_\star}\,(m_\star + 1)^{\frac{3}{2}\gamma_\star^+}\, T_{\mathrm{bc}}^{2/3}\right). \tag{28}$$

Let $m_\tau$ be the memory size associated to the bandit played at block time step $\tau$ by Algorithm 2. Let $m_{\min} = \min_j m_j$ and $m_{\max} = \max_j m_j$. Finally, let $L_{\min}$ and $L_{\max}$ the (partial) block length associated with $m_{\min}$ and $m_{\max}$. We have

$$\sum_{t=1}^{T} r_t^{\mathrm{bc}} \ge \sum_{\tau=1}^{T_{\mathrm{bc}}} \left(\widetilde{r}(\boldsymbol{a}_\tau^{\mathrm{bc}}) - m_\tau\,(m_\star + 1)^{\gamma_\star^+}\right) \ge \sum_{\tau=1}^{T_{\mathrm{bc}}} \widetilde{r}(\boldsymbol{a}_\tau^{\mathrm{bc}}) - m_{\max}\,(m_\star + 1)^{\gamma_\star^+}\, T_{\mathrm{bc}},$$

such that by Lemma 1 and (28) we obtain

$$\mathbb{E}\left[\frac{\min_\tau |\boldsymbol{a}_\tau|}{\max_\tau |\boldsymbol{a}_\tau|}\left(\widetilde{r}(\widetilde{\boldsymbol{a}})\,\frac{\sum_\tau |\boldsymbol{a}_\tau|}{|\widetilde{\boldsymbol{a}}|}\right) - \sum_{t=1}^{T} r_t^{\mathrm{bc}}\right]$$

$$\le m_{\max}\,(m_\star + 1)^{\gamma_\star^+}\, T_{\mathrm{bc}} + \min_\tau |\boldsymbol{a}_\tau|\, \widetilde{\mathcal{O}}\left(d\sqrt{m_\star}\,(m_\star + 1)^{\frac{3}{2}\gamma_\star^+}\, T_{\mathrm{bc}}^{2/3}\right),$$

$$\mathbb{E}\left[\frac{m_{\min} + L_{\min}}{m_{\max} + L_{\max}}\left(\frac{L_\star\,\mathrm{OPT}}{T}\,\frac{T}{m_\star + L_\star}\right) - \sum_{t=1}^{T} r_t^{\mathrm{bc}}\right]$$

$$\leq \frac{m_{\max}\,(m_\star + 1)^{\gamma_\star^+}\,T}{m_{\min} + L_{\min}} + (m_{\min} + L_{\min})^{1/3}\,\widetilde{\mathcal{O}}\Big(d\,\sqrt{m_\star}\,(m_\star + 1)^{\frac{3}{2}\gamma_\star^+}\,T^{2/3}\Big),$$

$$\mathbb{E}\left[\sqrt{\frac{m_{\min}}{m_{\max}}}\,\mathrm{OPT} - \sum_{t=1}^{T} r_t^{\mathrm{bc}}\right] \leq \frac{m_{\max}}{m_{\min}}\sqrt{d\,m_\star}\,(m_\star + 1)^{\gamma_\star^+}\,T^{3/4} + \widetilde{\mathcal{O}}\Big(d\,m_\star\,(m_\star + 1)^{\frac{3}{2}\gamma_\star^+}\,T^{3/4}\Big)$$

$$= \frac{m_{\max}}{m_{\min}}\,\widetilde{\mathcal{O}}\Big(d\,m_\star\,(m_\star + 1)^{\frac{3}{2}\gamma_\star^+}\,T^{3/4}\Big),$$

where we have used the fact that $m_{\min} + L_{\min} = \sqrt{m_{\min}/d}\,T^{1/4}$, and $m_{\max} + L_{\max} = \sqrt{m_{\max}/d}\,T^{1/4}$. Corollary 1 is obtained by setting $M = m_{\max}/m_{\min}$. □

# B   Bandit Combiner

In this section we show our adaptation of the numbers $C_j$ and target regrets $R_j$ for the Bandit Combiner algorithm Algorithm 2 which builds on Cutkosky et al. (2020). For $\mathtt{O3M}(m_j, \gamma_j)$, $j = 1, \ldots, N$, the numbers $C_j$ and target regrets $R_j$ are defined as

$$C_j = 4\left(\frac{4m_j}{d}\right)^{1/3}(m_j + 1)^{\gamma_j^+}\sqrt{d\ln\left(1 + \frac{T_{\mathrm{bc}}(m_j + L_j)(m_j + 1)^{2\gamma_j^+}}{d\lambda}\right)} \tag{29}$$
$$\left(\sqrt{\lambda} + \sqrt{\ln\left(\frac{1}{\delta}\right) + d\ln\left(1 + \frac{T_{\mathrm{bc}}(m_j + 1)^{2\gamma_j^+}}{d\lambda}\right)}\right),$$

$$R_j = C_j T_{\mathrm{bc}}^{\alpha_j} + \frac{(1 - \alpha_j)^{\frac{1-\alpha_j}{\alpha_j}}(1 + \alpha_j)^{\frac{1}{\alpha_j}}}{\alpha_j^{\frac{1-\alpha_j}{\alpha_j}}}C_j^{\frac{1}{\alpha_j}}T_{\mathrm{bc}}\eta_j^{\frac{1-\alpha_j}{\alpha_j}}$$
$$+ 1152(m_j + 1)^{2\gamma_j^+}\log(T_{\mathrm{bc}}^3 N/\delta)T_{\mathrm{bc}}\eta_j + \sum_{k \neq j}\frac{1}{\eta_k}.$$

Note that the form of the target regret $R_j$ slightly differs from the one presented in (Cutkosky et al., 2020, Corollary 2) due to the different range of the rewards. The algorithm, which is an adaptation of Bandit Combiner in Cutkosky et al. (2020), is summarized in Algorithm 2.

# C   Additional Experiments

We provide an additional experiment comparing the regrets of $\mathtt{O3M}$ and $\mathtt{OM\text{-}Block}$. In order to be able to plot the regret, we must know OPT which is hard to compute in general. Since in the rising scenario with an isotropic initialization OPT is oracle greedy, which is easy to compute, we present this experiment in a rising setting with $m = 1$ and $\gamma = 2$. We plot the regret of $\mathtt{O3M}$ and $\mathtt{OM\text{-}Block}$ against the number of time steps, measuring the performance at different time horizons and for different sizes of $L$ (where $L$ depends on $T$, see at the end of Section 3.2). Specifically, we instantiated $\mathtt{O3M}$ and $\mathtt{OM\text{-}Block}$ for increasing values of $L$, setting the horizon of each instance based on the equations in Theorem 1 and Proposition 4. Figure 3 shows how the dimension of $\widehat{\theta}$, which is $d$ for $\mathtt{O3M}$ and $d \times L$ for $\mathtt{OM\text{-}Block}$, has an actual impact on the performance since $\mathtt{O3M}$ outperforms $\mathtt{OM\text{-}Block}$. The code is written in Python and it is publicly available at the following GitHub repository: Linear Bandits with Memory.

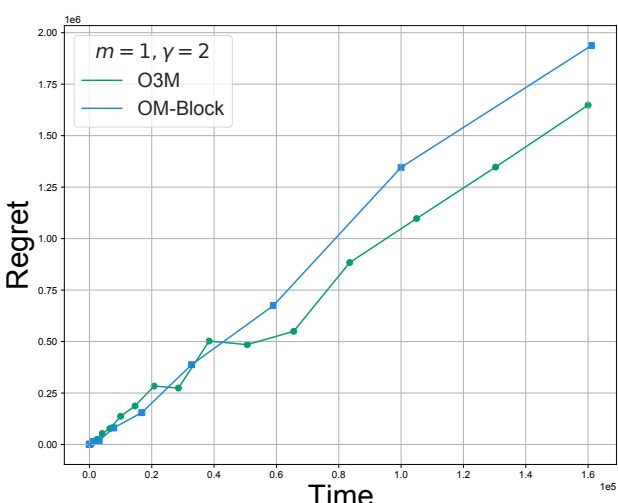

Figure 3: The regret of `O3M` and `OM-Block`. Each dot is a separate run where the value of $L$ is tuned to the corresponding horizon.

