# OpenReview forum: "Linear Bandits with Memory"
_TMLR — Accepted by TMLR_

### Review · Reviewer_uWLR · 2024-02-02

**Summary Of Contributions:**

In short, this paper is a linear bandit version of the rotting/rising bandit. The author assumes that the reward is non-stationary and depends on the previous decisions, especially on the memory matrix. Authors proved that in this sliding-window setting, the optimal policy can be tightly approximated by optimal cyclic policy, and using this idea they proposed a variant of OFUL, which minimizes the regret against the cyclic policies. This result holds for both rotting and rising bandit which is determined by the sign of the exponent $\gamma$. Not only that, using the model selection approach, they could extend their approach to all the positive and negative cases.

**Audience:**

Yes

**Broader Impact Concerns:**

There are no ethical concerns we need to worry about in this paper.

**Claims And Evidence:**

Yes

**Requested Changes:**

- I believe the authors should also justify the scale of the effect of the history. I mean, the authors assume $A_t = I + \sum a_s a_s^\top$, but it is also a natural question to ask the case $A_t = I + \alpha \sum a_s a_s^\top$ for some constant $\alpha$ and effect of this constant $\alpha$.

- I need more explanation to support why 'finite memory' is natural. I read their argument in Page 5 that using the well-known Eliptic potential lemma, it becomes a trivial problem. However, I believe the tendency of rotting ($\gamma <0$) and rising ($\gamma>0$) are quite different, as they demonstrated

- I need some valid explanation 2) It is not clear how the authors 'optimized' the sequence of batch actions. Though they mentioned they found the UCB indices by Gradient Ascent (Page 10, computational complexity), I think it would be great if authors also provided more explanation that computing block $a_{\tau}$ is easy (such as convex problem or sth? I am not super clear whether it is actually that easy problem).

- (Minor) What happens when the memory is not a sliding window, but a diminishing one? There are two representative ways of expressing the limitation of memory. One is the sliding window which the authors used in this paper, and the other is using discount factors to express the exponential decay of the memory. I think it would be great if the authors could also explain what happens with discount factors instead of this sliding window structure.

**Strengths And Weaknesses:**

Strength

- Novelty: It is kind of a natural process to think of an extension from the analysis of the K-armed bandits to linear bandits. However, I believe this paper well-defined and interpreted the concept of rising/rotting in the linear bandit setting. I think this is a topic someone should deal with someday, and the authors made a proper analysis.

- Quality: I want to value three main techniques.
1) $\langle a_t, A_{t-1} \theta^* \rangle = \langle A_{t-1} a_t, \theta^* \rangle$, so actually the researchers don't need to worry about the 'changing hidden parameter' $A_{t-1} \theta^*$, and one can take this rising/rotting environment as 'changing arm set' environment where the environment changes based on learner's choice and the learner fully understand its effect. I think it is a simple idea that researchers who have deeply studied this field may find it easily, but at least for me, it is a result that gives me a fresh intuition.
2) Instead of directly dealing with the 'true optimal sequence', they proved in Propositions 2 and 3 that one can use 'cyclic best action' as the alternative to the true optimal sequence since 'true optimal' and 'cyclic best' are tight. This is an interesting way to detour something difficult to compute. Though I haven't fully read the details of the proof, it is an interesting result for me if it is true.
3) It was surprising that even the learner knew $\theta^*$ beforehand, the 'oracle greedy policy' could have linear regret. It was very counter-intuitive. I haven't read the proof in detail, but if it is true then it is definitely worth to add points on this paper.

- Clarity: The authors tried to deliver the main ideas as straightforward as possible. Though it is not crystal clear to me yet, at least I could understand the mathematical structure they constructed for their analysis. Plus, I personally like Figure 1 they added to help readers understand better.


Weakness and Quality

- Novelty: (super minor) Using OFUL, the traditional approach for the linear bandit, is a kind of predictable approach.

- Clarity: I believe they need to explain some parts more clearly.

1) I hope they explain more about the ease of the 'infinite memory case.' I know they mentioned it on Page 5, but they also need to explain it on the rising bandit case too, I believe. It is not straightforward for me how rising will also be easy.

2) It is not clear how easy to find the best batch of actions $a_\tau$. Though they mentioned they found the UCB indices by Gradient Ascent (Page 10, computational complexity), I think it would be great if authors also provided more explanation that computing blocks $a_{\tau}$ is easy (such as a convex problem or sth? I am not super clear whether it is that easy problem).

---

> ### Author Response · Authors · 2024-03-21
>
> > I believe the authors should also justify the scale of the effect of the history. I mean, the authors assume $A_t = I + \sum_{s=1}^m a_s a_s^{\top} $ but it is also a natural question to ask the case $A_t = I + \alpha \sum_{s=1}^m a_s a_s^{\top}$ for some constant $\alpha$ and effect of this constant $\alpha$.
>
> Thank you for this remark.
> Although the model proposed by the reviewer can certainly be considered, we preferred not to introduce an extra hyperparameter $\alpha$ for the following reasons:
> * First, we highlight that the "scale of the effect of the history" is already taken into account by the exponent $\gamma$, and that the model proposed by the reviewer is essentially a first-order approximation of ours. Indeed, consider the action space to be the standard basis in $\mathbb{R}^d$, i.e., $\mathcal{A} = \{e_k \colon k \le d\}$. The memory matrix then becomes $A_t = \text{diag}\big((1+n_k)^\gamma\big)$, where $n_k = \sum_{s=1}^m \mathbb{I}\{a_{t-s} = e_k\}$ is the number of times action $e_k$ has been played in the past $m$ rounds. A first order approximation gives $A_t \approx \text{diag}\big(1 + \gamma\,n_k\big)$, which coincides with the model proposed by the reviewer (for $\alpha = \gamma$).
>
> * Second, we recall that the definition of the memory matrix given in Equation (2) is: $A(a_1, \dots, a_m) = \left(A_0 + \sum_{s=1}^m a_s a_s^{\top}\right)^\gamma$. Hence, although we choose $A_0 = I_d$ in most scenarios, if $\alpha$ is known our model perfectly allows to set $A_0 = \frac{1}{\alpha} I_d$, i.e., $A(a_1, \dots, a_m) = \left(\frac{1}{\alpha}I_d + \sum_{s=1}^m a_s a_s^{\top}\right)^{\gamma}$, which matches the proposition made by the reviewer up to the reparameterization $\alpha \leftrightarrow 1/\alpha$. If $\alpha$ is unknown, however, learning this extra hyperparameter would degrade the regret bound with model selection (Corollary 1).
> \
> \
> $~$
>
> > I need more explanation to support why 'finite memory' is natural. I read their argument in Page 5 that using the well-known elliptical potential lemma, it becomes a trivial problem. However, I believe the tendency of rotting ($\gamma < 0$) and rising ($\gamma > 0$) are quite different, as they demonstrated.
>
> Please note that we also justify the choice of a finite memory $m$ from a practical perspective. As we mentioned, in the song recommendation problem we study, it is not realistic to assume that a song affects the user's preferences indefinitely. Moreover, we showed in Example 2 how assuming an infinite memory $m$ is to trivialize the problem in the rotting setting. We did not mention the rising case, as we believe that exhibiting one counterexample where an infinite $m$ makes the problem trivial is a valid justification for not adopting this modeling prior.
> \
> \
> $~$
>
> > It is not clear how the authors 'optimized' the sequence of batch actions. Though they mentioned they found the UCB indices by Gradient Ascent (Page 10, computational complexity), I think it would be great if authors also provided more explanation that computing block $a_{\tau}$ is easy (such as convex problem or sth? I am not super clear whether it is actually that easy problem).
>
> Thank you for raising this point. Indeed, maximizing the UCBs is a hard problem when the action space is infinite, that might be non-convex in general. In that respect, the theoretical guarantees we provide in Theorem 1 hold whenever the learner has access to some oracle that returns the exact UCB maximizer, as traditionally assumed in the literature, see e.g., [1]. Conversely, note that the practical implementation of O3M still satisfies Theorem 1, but for a slightly weaker version of the regret where the "best block" is understood as the one returned by the approximated oracle used in O3M (i.e., our Gradient Ascent solver). See Section 9 in [1] for a similar discussion. We added this discussion to Section 4.
> \
> \
> [1] B. Kveton, Z. Wen, A. Ashkan, and C. Szepesvari. *Tight regret bounds for stochastic combinatorial semi-bandits.* In Artificial Intelligence and Statistics, pages 535–543. PMLR.
> \
> \
> $~$
> > (Minor) What happens when the memory is not a sliding window, but a diminishing one? There are two representative ways of expressing the limitation of memory. One is the sliding window which the authors used in this paper, and the other is using discount factors to express the exponential decay of the memory. I think it would be great if the authors could also explain what happens with discount factors instead of this sliding window structure.
>
> Using discount factors is indeed another standard way to model the impact of past actions. We highlight however that this would considerably change the analysis (e.g., the approximation result of Proposition 2), and goes beyond the scope of the present paper. Furthermore, note that the discount factor is usually assumed to be known, while our sliding window model makes it possible to learn $m$. We added this remark as a footnote on Page 3.

---

### Review · Reviewer_CVyS · 2024-02-10

**Summary Of Contributions:**

The author propose a new linear bandits framwork LBM where the reward at time t is affected by all the actions within [t-m, t] blocks. This frameworks generalized the stationary linear bandits, rising and rotting rested bandits and cyclic bandits. Under this framework, the author first proposes a variant of OFUL algortihm with the known-parameter m. Then the author propose a parameter-free algortihm by using a meta-algorihtm to select the right model with an additional $M\sqrt{d(m+1)^{1+\gamma}}$ multiplicative term in the final bound.

**Audience:**

Yes

**Claims And Evidence:**

Yes

**Requested Changes:**

I think it is a little unclear for me on the over-optimistic part (O3M). I can understand it has the same proofs as the OM. Are you mentioned this because it is more easy to implement? or has better emprirical results ?

**Strengths And Weaknesses:**

Strength:
1. The author proposes a novel bandits framework that can generalize several structured nonstationry bandits problem including bandits, rising and rotting rested bandits and cyclic bandits, and it can also generalize a special case of MDP problem.
2. I also like the way they explain this problem step by step. It is quite clear. For example, in Section 3.1. They discuss the influence of initial m conditions. In Section 3.1,3.2, they give a clear explaination on the trade-off on L between long tern plainning and infrequent updating.
3. Their results, although may not be optimal, is quite compelete as the first paper proposing this.

Weakness:
1. It is unclear to me wether this solution is optimal, seems a very direct solutions.
2. It might be interesting to talk about some instance-dependent bound under this setting
But overall I think the results are enough as the first paper under this framework.

---

> ### Author Response · Authors · 2024-03-21
>
> >It is unclear to me whether this solution is optimal, seems a very direct solution.
>
> As we discussed on page 8, below Theorem 1, the analysis we present is tight for some special cases. However, finding a matching instance-based lower bound is a nontrivial problem, hardly solved in nonstationary bandit settings.
> \
> \
> $~$
>
> >I think it is a little unclear for me on the over-optimistic part (O3M). I can understand it has the same proofs as the OM. Are you mentioned this because it is more easy to implement? or has better empirical results?
>
> Indeed, we mainly introduced O3M because of its better empirical performances. As discussed in Remark3, we attribute them to the fact that the confidence set it is built upon is more optimistic. We added this remark in the revision.

---

### Review · Reviewer_WoTz · 2024-03-14

**Summary Of Contributions:**

This paper studies a bandit problem where the observed (scalar) outcome of a given round depends on the latest $m+1$ actions. The authors propose an OFUL-based algorithm for this setting and provide an upper bound on the regret thereof against the optimal sequence of actions.

**Audience:**

Yes

**Broader Impact Concerns:**

Not applicable.

**Claims And Evidence:**

Yes

**Requested Changes:**

Please see the "Strengths And Weaknesses" section.

**Strengths And Weaknesses:**

Why is the model in eq (1) and (2) a reasonable or justified way of doing things? Why model memory via a covariance matrix? Why not simply use a linear model that depends on the latest $m$ actions? E.g., $y_t = \sum_{i=0}^{m-1}\theta_{i}^\ast a_{t-i} + \eta_t$, which can be recast as a standard linear model ($m=1$) in a higher dimensional space. One can then simply invoke the technical machinery already developed for linear bandits to essentially "solve" this richer setting.

Even if there are computational/statistical benefits to modeling memory via a covariance matrix, I do not see how it is a natural or intuitive way to do it compared to the simple alternative described above? The paper currently does not provide any background/motivation on this, and I think the authors need to better explain why their model is reasonable/justified.

In addition to the above, the "rotting/rising" settings need better explanation and motivation. Currently, they are simply passed off as settings with a positive or negative $\gamma$. Further, this model is, in fact, not linear as the outcome depends non-linearly on past and present actions unless $\gamma=0$.

The current version of the paper lacks a coherent example/motivation to ground the model. The mathematics is fine, but the reader is likely to walk away with the impression that the paper analyzes and provides results for an esoteric model that is a strict generalization of linear bandits and a partial generalization of some other models. The current positioning and motivation leaves much to be desired.

---

> ### Author Response · Authors · 2024-03-21
>
> >Why is the model in eq (1) and (2) a reasonable or justified way of doing things? Why model memory via a covariance matrix? Why not simply use a linear model that depends on the latest m actions? E.g., $y_t= \sum_{i=0}^{m-1} \theta^*_i a\_{t-i} + \eta_t$, which can be recast as a standard linear model ($m=1$) in a higher dimensional space. One can then simply invoke the technical machinery already developed for linear bandits to essentially "solve" this richer setting.
>
> >Even if there are computational/statistical benefits to modeling memory via a covariance matrix, I do not see how it is a natural or intuitive way to do it compared to the simple alternative described above? The paper currently does not provide any background/motivation on this, and I think the authors need to better explain why their model is reasonable/justified.
>
> >In addition to the above, the "rotting/rising" settings need better explanation and motivation. Currently, they are simply passed off as settings with a positive or negative $\gamma$. Further, this model is, in fact, not linear as the outcome depends non-linearly on past and present actions unless $\gamma = 0$.
>
> As mentioned in the introduction, our goal is to extend the rotting/rising models previously studied in the $K$-armed setting only (i.e., where each arm rots/rises independently of the other arms) to the linear bandit framework. The key to this extension is capturing cross-arm effects, as motivated in the rock vs. folk music example at the bottom of Page 1. The $K$-armed setting is just a special case of the linear setting, in which the $K$ actions form an orthonormal basis of the space, and the covariance matrix becomes diagonal (i.e., no correlation among arms). The values on the diagonal of the covariance matrix are then equal to the number of times each arm has been pulled so far. When all actions in the linear space are available, and not just a basis of the space like in the $K$-armed case, the covariance matrix is not diagonal anymore, and its spectrum reveals the directions corresponding to the actions played most often in the past, see Figure 1. This is the property we implicitly use in our extension of the rotting/rising model, and the main motivation for using the covariance matrix. It is unclear to us how your linear model could be used for the same purpose. Regarding the motivation of the rotting/rising model, we mostly refer the reader to previous work, as we believe these are quite established scenarios in the bandit literature.

---

### Author Response · Authors · 2024-03-21

Dear reviewers and action editor,
\
\
We thank the reviewers for their feedback. We edited our manuscript (in blue) to address their comments, see the revision. We also provided below detailed answers.
We remain available for further interaction.
\
\
Best,

the authors

---

### Author Response · Authors · 2024-06-04
**Camera Ready Revision**

Dear reviewers and action editor,

This message is to thank you once again for your valuable feedback.

We edited our manuscript to address your comments, focusing in particular on expanding the discussion on finite memory $m$, see Example 2 (pages 4-5).

Best,

the authors

---

### Decision · Action_Editor_mfEn · 2024-05-04

**Recommendation:** Accept with minor revision

**Comment:**

The paper introduces a new nonstationary linear bandit model, to capture situations where rewards depend on the learner’s past actions in a fixed-length sliding window. This model (approximately) recovers a few previously bandit problems. Weaknesses include a matching lower bound, and a somewhat limited class of polynomial nonstationary reward functions. Overall, reviewers found the work to be technically sound and contribute meaningfully to the bandit literature. Some of the technical results are also interesting, such as using the cyclic policy as a reference as part of the analysis.

Please use the reviews to revise the paper, including strengthening the motivation, among others. In addition, can you elaborate further why finite m avoid trivializing the theory? The current argument relies on gamma to be sufficiently small.If gamma is large, does an infinite m still trivialize the theory?

**Audience:**

The paper is interesting to the subcommunities of bandit / RL.

**Claims And Evidence:**

The paper introduces a new nonstationary linear bandit model, to capture situations where rewards depend on the learner’s past actions in a fixed-length sliding window. Most technical results are theoretical, justified by proofs. The paper also has numerical experiments on synthetic problems. Claims are supported.